# Towards Efficient Online Tuning of VLM Agents via Counterfactual Soft Reinforcement Learning

**Lang Feng**[1]  **Weihao Tan**[1]  **Zhiyi Lyu**[1]  **Longtao Zheng**[1]  **Haiyang Xu**[2]  **Ming Yan**[2]  **Fei Huang**[2]  **Bo An**[1]

## Abstract

Online fine-tuning vision-language model (VLM) agents with reinforcement learning (RL) has shown promise for equipping agents with multi-step, goal-oriented capabilities in dynamic environments. However, their open-ended textual action space and non-end-to-end nature of action generation present significant challenges to effective online exploration in RL, e.g., explosion of the exploration space. We propose a novel online fine-tuning method, Counterfactual Soft Reinforcement Learning (CoSo), better suited to the textual output space of VLM agents. Compared to prior methods that assign uniform uncertainty to all tokens, CoSo leverages counterfactual reasoning to dynamically assess the causal influence of individual tokens on post-processed actions. By prioritizing the exploration of action-critical tokens while reducing the impact of semantically redundant or low-impact tokens, CoSo enables a more targeted and efficient online rollout process. We provide theoretical analysis proving CoSo's convergence and policy improvement guarantees, and extensive empirical evaluations supporting CoSo's effectiveness. Our results across a diverse set of agent tasks, including Android device control, card gaming, and embodied AI, highlight its remarkable ability to enhance exploration efficiency and deliver consistent performance gains. The code is available at https://github.com/langfengQ/CoSo.

## 1. Introduction

Large Vision-Language Models (VLMs) like Qwen-VL (Bai et al., 2025b), Gemini (Team et al., 2023), and GPT-4o (Achiam et al., 2023), are increasingly employed as decision-making agents for autonomous tasks in both physical or virtual environments, with applications in device control (Wang et al., 2024b; Hu et al., 2024; Furuta et al., 2024), gaming (Wang et al., 2024a; Tan et al., 2024b), robotic control (Zhao et al., 2023; Zitkovich et al., 2023), and autonomous driving (Tian et al., 2024).

Despite these advantages, adapting VLM agents to dynamic environments remains a challenge due to the limited task-specific capabilities of the base models (Zeng et al., 2023; Chen et al., 2023). Recent advances (Bai et al., 2024; Zhai et al., 2024; Wen et al., 2024) leverage reinforcement learning (RL) (Sutton & Barto, 2018) for online fine-tuning, which has shown particular promise. RL enables VLM agents to iteratively interact with online environments and optimize for multi-step, goal-directed objectives. This aligns naturally with the requirements of autonomous systems. For instance, in Android device controls, agents manage multi-step workflows like launching the Amazon app, locating specific items, and completing purchases.

However, the RL fine-tuning of VLM agents presents unique challenges for the *online exploration*. Two primary factors contribute to these difficulties. First, VLM agents operate in a text-based action space, where the open-ended text often includes thoughts, planning, and actions (Zhang et al., 2023; Niu et al., 2024). Unlike traditional RL, which enables vectorized action sampling via softmax distributions, the sampling space for VLM agents' textual outputs scales exponentially with the vocabulary size and sequence length. This immense policy space significantly complicates effective exploration. Second, action generation in VLM agents is not end-to-end. Generated textual outputs (utterances) need to be post-processed into executable actions via parsing functions that match the environment's API calls (Zhai et al., 2024; Bai et al., 2024). It introduces a misalignment between exploring textual utterances and exploring final parsed actions. Many tokens in the utterances, particularly those with fixed formats or semantically redundant elements, do not contribute meaningful changes to the post-processed actions. As a result, naive RL exploration becomes inefficient and struggles to drive meaningful progress.

In this study, we propose a novel RL fine-tuning method, better suited to the online exploration of VLM agents' tex-

---

[1]Nanyang Technological University [2]Alibaba Group. Correspondence to: Bo An <boan@ntu.edu.sg>.

*Proceedings of the 42nd International Conference on Machine Learning*, Vancouver, Canada. PMLR 267, 2025. Copyright 2025 by the author(s).

tual action space. The core insight of our approach is that the influence of tokens on the parsed action varies, with a small subset of action-critical tokens decisively shaping the final outcome. In contrast to naive RL training that treats all tokens as equally important, our proposal, Counterfactual Soft Reinforcement Learning (CoSo), analyzes the causal contributions of VLM-generated tokens to the actions. By leveraging counterfactual reasoning to compute causal weights for tokens, CoSo introduces a causal-weighted entropy-regularized optimization built upon soft RL (Haarnoja et al., 2017). This mechanism performs more targeted and efficient exploration, focusing on the causally significant tokens that induce meaningful variations in actions rather than random trial-and-error. As such, CoSo significantly reduces the redundant exploration space and enables the exploration of open-ended utterances to align more closely with the exploration of final parsed actions.

We provide a theoretical justification of CoSo's convergence and effectiveness. In our analysis, CoSo establishes guarantees for both policy evaluation and policy improvement. As CoSo provides a highly general objective, it can be flexibly adapted to the more specific RL objective for agents' fine-tuning. In our empirical evaluations, we extend CoSo to two RL fine-tuning objectives: an AWR-based approach (Bai et al., 2024) and a PPO-based approach (Zhai et al., 2024), subjecting VLM agents across various autonomous tasks, including Android device control (Rawles et al., 2024), card gaming (Brockman, 2016), and embodied AI (Shridhar et al., 2021). Our results demonstrate its impressive ability to enhance the efficiency and effectiveness of online fine-tuning. Over existing RL frameworks, CoSo-reinforced agents achieve consistent performance improvements, with average gains of 12.3% in Android device control, 9.3% in card gaming, and 16.7% in embodied AI tasks.

## 2. Related Work

**LLM/VLM as agents.** Recently, there has been a surge of interest in using large language models (LLMs) or VLMs as goal-driven agents in various domains like code (Zhang et al., 2024b), device control (Gur et al., 2024), gaming (Wang et al., 2024a), and robotic control (Zitkovich et al., 2023). One line of works focuses on test-time scaling by incorporating sophisticated modules (Wang et al., 2024b; Tan et al., 2024b), prompting techniques (Yao et al., 2023; Shinn et al., 2024), or external tools (Schick et al., 2023; Xie et al., 2024; Zhang et al., 2024a) with frozen models like GPT-4 (Achiam et al., 2023). Another line explores improving the task adaptability of agents by fine-tuning the base models with Supervised Fine-Tuning (SFT) (Zhang & Zhang, 2024) or RL (Carta et al., 2023; Tan et al., 2024a). These fine-tuned models often exhibit improved generalization and robustness, making them better suited for special-

ized tasks.

**Reinforcement learning for LLM/VLM training.** RL has become a cornerstone for improving the capabilities of LLMs and VLMs. A prominent application is Reinforcement Learning with Human Feedback (RLHF) (Ouyang et al., 2022; Rafailov et al., 2024), which aligns models with human preferences and ethical guidelines by iteratively refining their outputs. Also, RL has been applied to enhance reasoning capabilities like DeepSeek-R1 (Liu et al., 2024a; Guo et al., 2025), enabling models to perform complex multi-step reasoning and logical deductions. In the LLM/VLM agent domain, works like (Wen et al., 2024; Zhai et al., 2024; Bai et al., 2024; 2025a; Wang et al., 2025) have employed RL for online fine-tuning, creating agents that can adapt and learn from dynamic environments. Yet, they rely on classic trial-and-error for online exploration, which can be inefficient in open-ended text action spaces.

**Exploration in RL.** Effective exploration remains a key challenge in RL, particularly in complex environments. Intrinsic motivation methods (Bellemare et al., 2016; Pathak et al., 2017; Burda et al., 2018) address this challenge by introducing auxiliary reward signals that drive agents to explore novel states. On the other hand, entropy-based methods (Ziebart, 2010; Haarnoja et al., 2017; 2018; Ji et al., 2024) encourage diverse policy behaviors by increasing decision uncertainty. While effective in balancing exploration and exploitation for RL agents operating in vectorized action spaces, such uncertainty on textual action could inadvertently complicate exploration for VLM agents. Our method reduces unnecessary complexity by prioritizing impactful tokens, enabling efficient online fine-tuning for VLM agents.

## 3. Preliminaries

**Problem Settings and MDP.** The task-completion process of a VLM agent involves completing a series of actions in response to a user instruction in natural language to achieve a specified goal. The entire process of agent-environment interactions can be modeled as a finite-horizon Markov Decision Process (MDP) (Sutton & Barto, 2018), defined by the tuple $(\mathcal{S}, \mathcal{A}, P, r, \gamma)$. The state space $\mathcal{S}$ combines visual and textual modalities, defined as $\mathcal{S} = \mathcal{O} \times \mathcal{V}^m$, where $\mathcal{O}$ denotes the RGB image space, $\mathcal{V}$ denotes the vocabulary (token) space, and $m$ denotes the maximum token length of the input sequence. $\mathcal{A}$ is the action space, consisting of executable actions translated from the VLM-generated utterances via a deterministic parsing (post-processing) function $f^{\text{parse}} : \mathcal{V}^n \rightarrow \mathcal{A}$ with $n$ denoting the maximum output token length. The transition probability function $P : \mathcal{S} \times \mathcal{A} \times \mathcal{S} \rightarrow [0, 1]$ defines the likelihood of transitioning between states given an action. $r : \mathcal{S} \times \mathcal{A} \rightarrow [r_{\min}, r_{\max}]$ is the reward function and $\gamma$ is the discounted factor. At each time step $t$, the VLM agent occupies a state $\boldsymbol{s}_t = (\boldsymbol{o}_t, \boldsymbol{x}_t)$,

where $\boldsymbol{o}_t \in \mathcal{O}$ and $\boldsymbol{x}_t \in \mathcal{V}^m$, and then generates an action $\boldsymbol{a}_t = f^{\text{parse}}(\boldsymbol{y}_t)$ with the utterance $\boldsymbol{y}_t \in \mathcal{V}^n$. This action induces a transition to the next state $\boldsymbol{s}_{t+1} = P(\boldsymbol{s}_t, \boldsymbol{a}_t)$, and the agent receives a reward $r(\boldsymbol{s}_t, \boldsymbol{a}_t)$.

**RL fine-tuning for VLM agents.** VLM with parameter $\theta$ is used as the policy of agent $\pi_\theta(\boldsymbol{a}|\boldsymbol{s})$, i.e., given a environment state $\boldsymbol{s}$, output an action $\boldsymbol{a}$. To use RL to online fine-tune the VLM agent, the process typically follows two key steps: (1) *Online exploration*: the VLM agent interacts with the environment to sample rollout data based on its current policy. (2) *Exploitation*: the collected rollout data is then used to optimize the VLM policy using RL algorithms like PPO (Schulman et al., 2017). The general objective of RL fine-tuning is to maximize the expected cumulative reward: $\max_\pi \sum_t \mathbb{E}_{(\boldsymbol{s}_t, \boldsymbol{a}_t) \sim \rho^\pi} [\gamma^t r(\boldsymbol{s}_t, \boldsymbol{a}_t)]$, where $\rho^\pi$ is the state-action visitation distribution under policy $\pi$. We omit the subscript of $\pi_\theta$ for convenience.

**Online exploration with entropy regularization.** Efficient data sampling during the online exploration phase is crucial for optimizing the VLM agent's performance. To this end, RL often incorporates an entropy regularization term to encourage exploration, yielding the soft RL optimization, the objective of which can be expressed as:

$$\max_\pi \sum_t \mathbb{E}_{(\boldsymbol{s}_t, \boldsymbol{a}_t) \sim \rho^\pi} \left[ \gamma^t (r(\boldsymbol{s}_t, \boldsymbol{a}_t) + \alpha \mathcal{H}(\pi(\cdot|\boldsymbol{s}_t))) \right], \quad (1)$$

where $\alpha \geq 0$ is a weighting coefficient, and $\mathcal{H}(\pi(\cdot|\boldsymbol{s}))$ denotes the entropy of the policy.

Note that, unlike traditional RL agents, the policy of the VLM agent is not an end-to-end function but rather a multi-step, segmented process: (**1**) The VLM first autoregressively generates an output utterance $\boldsymbol{y} = (y^1, \ldots, y^n)$ via $\pi(y^i|\boldsymbol{y}^{1:i-1}, \boldsymbol{s})$, where $\boldsymbol{y}$ typically includes elements such as chain-of-thought (CoT) (Wei et al., 2022) and textual actions. (**2**) The sequence $\boldsymbol{y}$ is then transformed into an executable action $\boldsymbol{a}$ through a deterministic parsing function $\boldsymbol{a} = f^{\text{parse}}(\boldsymbol{y})$. In this context, the entropy term $\mathcal{H}(\pi(\cdot|\boldsymbol{s}))$ over action $\boldsymbol{a}$ in Equation (1) is decomposed into the sum of the conditional entropies over the tokens $y^1, \ldots, y^n$:

$$\mathcal{H}(\pi(\cdot|\boldsymbol{s})) = \sum_{i=1}^n \mathcal{H}(y^i|\boldsymbol{y}^{1:i-1}), \quad (2)$$

where $\mathcal{H}(y^i|\boldsymbol{y}^{1:i-1})$ denotes the conditional entropy of $y^i$ given $\boldsymbol{y}^{1:i-1}$. We provide the proof in Appendix A.1. This feature introduces a significant challenge for RL in training VLM agents: *explosion of the exploration space*. For instance, we consider a VLM agent applied to Android mobile control tasks. Typically, there are six actions available: `Type`, `Click`, `Slide`, `Home`, `Back`, and `Enter`. In traditional RL, the action sampling space at each state is $|\mathcal{A}| = 6$. However, in the case of VLM agents, the exploration space scales exponentially with sequence length and

vocabulary size: $|\mathcal{V}|^n = 32,100^{100}$ (assuming a vocabulary size of 32,100 and a sequence length of 100 tokens). Notably, this space includes many tokens with fixed formats or semantic redundancies, which contribute little to meaningful action exploration. This immense combinatorial complexity significantly impedes exploration efficiency during the online fine-tuning of VLM agents, posing a fundamental challenge for RL online optimization.

# 4. Training VLM Agents with CoSo

While VLM policies offer the advantage of executing intermediate reasoning and planning steps, enabling efficient cognitive steps that lead to final decisions, their text-based outputs cause the online exploration highly inefficient when using RL for fine-tuning. This problem grows more pronounced as the sequence length increases. To address this, we introduce our Counterfactual Soft Reinforcement Learning (CoSo) in this section. CoSo innovatively introduces a counterfactual structure to analyze the causal relationships between the VLM-generated tokens $\boldsymbol{y} = (y^1, \ldots, y^n)$ and the post-processed action $\boldsymbol{a}$. Built upon this, CoSo introduces a causal-weighted entropy-regularized optimization based on soft RL (Haarnoja et al., 2017). This process enhances agents' exploration of action-critical tokens ($< 10\%$ in our experiments) while minimizing perturbations to low-impact tokens ($> 80\%$). Considering our previous example, CoSo can effectively narrow the sampling space to approximately 10 key tokens out of 100 total tokens, significantly improving the efficiency of exploration by $32,100^{90}$-fold.

In the following sections, we will delve into the process of deriving token-to-action causality through counterfactual reasoning, outline the causal-weighted entropy-regularized objective, provide a theoretical analysis of CoSo's effectiveness, and finally present its practical implementation.

## 4.1. Token-to-Action Causality

In VLM-powered agents, the relationship between the generated utterance $\boldsymbol{y}$ and the resulting parsed action $\boldsymbol{a}$ can be viewed as a structured causal process. Each token in the sequence $\boldsymbol{y}$ contributes to the action $\boldsymbol{a}$ via $f^{\text{parse}}$, yet not all tokens $y^1, \ldots, y^n$ exert the same level of influence on $\boldsymbol{a}$. Some tokens are pivotal in determining the action, while others have negligible or no causal impact. To formalize this relationship, we explicitly encode the token-to-action causal structure into the parsing function as follows:

$$\boldsymbol{a} = f^{\text{parse}}(\mathcal{B}_{\boldsymbol{y} \to \boldsymbol{a}} \odot \boldsymbol{y}), \quad (3)$$

where $\odot$ denotes the element-wise product, $\mathcal{B}_{\boldsymbol{y} \to \boldsymbol{a}} \in \mathbb{R}^{n \times 1}$ is a vector that represents the graph structure encoding both causal directions and causal effects. Specifically, $\mathcal{B}_{\boldsymbol{y} \to \boldsymbol{a}}^i = 0$ indicates the absence of a causal edge between token $\boldsymbol{y}^i$ and action $\boldsymbol{a}$, while $\mathcal{B}_{\boldsymbol{y} \to \boldsymbol{a}}^i = c$ (where $c > 0$) implies that $\boldsymbol{y}^i$

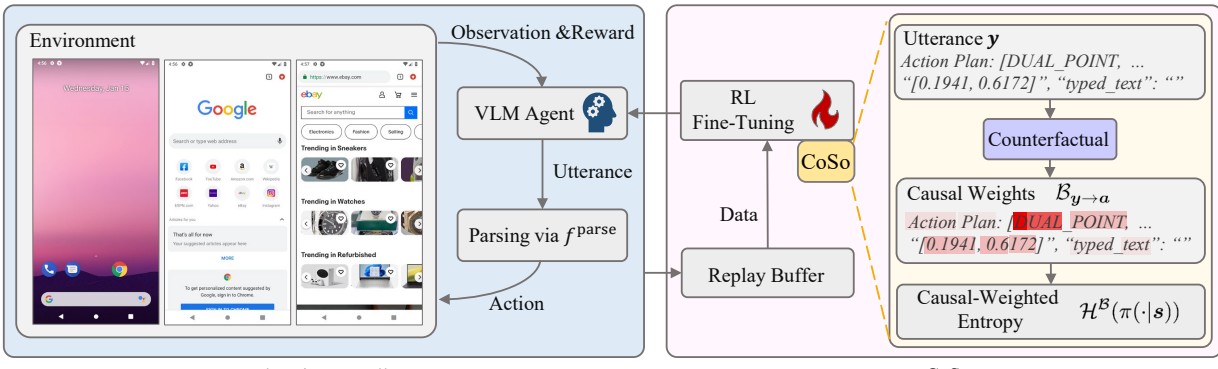

*Figure 1.* Overview of the CoSo framework for online fine-tuning VLM agents. The process includes two modules: (1) *Exploration and rollout*, where the VLM agent interacts with the environment to generate online data; (2) *CoSo*, which fine-tunes the VLM for multi-step, goal-directed objectives while promoting efficient exploration of action-critical tokens. Darker red indicates higher causal importance.

has a causal influence on $\boldsymbol{a}$ with effect magnitude $c$.

## 4.2. Counterfactual Reasoning

To determine the causal weights $\mathcal{B}_{\boldsymbol{y} \to \boldsymbol{a}}$, we leverage counterfactual reasoning, which facilitates the analysis of "what-if" scenarios by considering hypothetical interventions. To formalize this, we begin by introducing the basic assumption for conducting counterfactual analysis.

**Assumption 4.1** (Structural Causal Model). We assume that the relationship between the tokens $\boldsymbol{y}^i$ and the action $\boldsymbol{a}$ can be captured using a structural causal model (SCM) defined by $\boldsymbol{a} = f^{\mathrm{SCM}}(\boldsymbol{y}, \epsilon_{\boldsymbol{a}})$, where $\epsilon_{\boldsymbol{a}}$ represents an exogenous independent and identically distributed (i.i.d.) noise term.

Then, we compute the causal weight of each token $\boldsymbol{y}^i$ by measuring the sensitivity of the parsed action $\boldsymbol{a}$ to counterfactual interventions on $\boldsymbol{y}^i$. Specifically, for each token $\boldsymbol{y}^i$, we compare the action $\boldsymbol{a}$ resulting from the original sequence $\boldsymbol{y}$ with the action resulting from a counterfactual sequence in which $\boldsymbol{y}^i$ is replaced with a nullified value $\boldsymbol{y}^i_{\mathrm{null}}$ while keeping all other tokens unchanged. Formally, this process is defined as:

$$\mathcal{B}_{\boldsymbol{y} \to \boldsymbol{a}}^i = \mathbb{E}_{\boldsymbol{y}^{-i}}\left[ D\left(f^{\mathrm{SCM}}(\boldsymbol{y}, \epsilon_{\boldsymbol{a}}), f^{\mathrm{SCM}}(\boldsymbol{y}^{-i} \cup \boldsymbol{y}^i_{\mathrm{null}}, \epsilon_{\boldsymbol{a}})\right)\right], \tag{4}$$

where $D(\cdot, \cdot)$ is a distance metric that quantifies the change in the action $\boldsymbol{a}$ resulting from the intervention. $\boldsymbol{y}^{-i}$ denotes the sequence excluding token $\boldsymbol{y}^i$.

In practice, the effect of intervening a single token $\boldsymbol{y}^i$ on $f^{\mathrm{SCM}}$ can be subtle, making it challenging to observe an absolute shift on parsed action $\boldsymbol{a}$. To capture the fine-grained effects, we analyze changes in the output likelihood of the action rather than absolute shifts in outcomes. Specifically, we define $f^{\mathrm{SCM}}(\boldsymbol{y}, \epsilon_{\boldsymbol{a}})$ using a probabilistic model followed by an `argmax` operator, which can be expressed as:

$$f^{\mathrm{SCM}}(\boldsymbol{y}, \epsilon_{\boldsymbol{a}}) = \arg\max_{\boldsymbol{a}} \mathbb{P}(\boldsymbol{a}|\boldsymbol{y}, \epsilon_{\boldsymbol{a}}). \tag{5}$$

This probabilistic formulation is straightforward to implement, for instance, by using a surrogate model with a `softmax` output layer to instantiate the SCM. Building on this, we define the distance metric $D(\cdot, \cdot)$ as the likelihood difference and thereby the causal weight $\mathcal{B}_{\boldsymbol{y} \to \boldsymbol{a}}^i$ for each token $\boldsymbol{y}^i$ is given by

$$\mathcal{B}_{\boldsymbol{y} \to \boldsymbol{a}}^i = \left| \mathbb{P}(\boldsymbol{a}|\boldsymbol{y}, \epsilon_{\boldsymbol{a}}) - \mathbb{P}(\boldsymbol{a}|\boldsymbol{y}^{-i} \cup \boldsymbol{y}^i_{\mathrm{null}}, \epsilon_{\boldsymbol{a}}) \right|. \tag{6}$$

This approach enables a more sensitive measurement of the causal influence each token $\boldsymbol{y}^i$ exerts on the action $\boldsymbol{a}$, thereby facilitating a nuanced analysis of the interactions between VLM-generated utterances and the post-processed actions under the SCM framework.

## 4.3. Counterfactual Soft RL

Based on the causal weights $\mathcal{B}_{\boldsymbol{y} \to \boldsymbol{a}}$, we next propose our counterfactual soft RL, a novel approach that explicitly leverages the identified per-token causal importance to guide the maximization entropy policy optimization of VLM agents. The central idea of CoSo is to prioritize exploration around action-critical tokens in the generated utterance $\boldsymbol{y}$ instead of aggregating uniform uncertainty across all tokens. Specifically, we introduce a causal-weighted entropy term into the soft RL objective (Equation (1)), yielding:

$$\max_{\pi} \sum_t \mathbb{E}_{(\boldsymbol{s}_t, \boldsymbol{a}_t) \sim \rho^{\pi}}\left[\gamma^t\left(r(\boldsymbol{s}_t, \boldsymbol{a}_t) + \alpha \mathcal{H}^{\mathcal{B}}(\pi(\cdot|\boldsymbol{s}_t))\right)\right],$$
$$\text{where } \mathcal{H}^{\mathcal{B}}(\pi(\cdot|\boldsymbol{s})) = \sum_{i=1}^n \mathcal{B}_{\boldsymbol{y} \to \boldsymbol{a}}^i \cdot \mathcal{H}(y^i|\boldsymbol{y}^{1:i-1}) \tag{7}$$

Intuitively, if token $y^i$ is causally important ($\mathcal{B}_{\boldsymbol{y} \to \boldsymbol{a}}^i$ is high), we prioritize exploring its possibilities since their changes can effectively lead to the changes of the final parsed action. Conversely, tokens that do not lead to meaningful changes in $\boldsymbol{a}$ are assigned lower entropy weight, reducing the incentive to explore them. By doing so, CoSo achieves a more efficient exploration process, targeting the space with a higher likelihood of meaningful impact on action.

**Algorithm 1** Online fine-tuning VLM agents with CoSo
___
1: **Require:** VLM with parameters $\theta$, SCM with parameters $\phi$, parsing function $f^{\text{parse}}$, and environment env.
2: **for** $k = 0, \ldots, K-1$ **do**
3:    Initialize the environment $s_0 = $ env.reset()
4:    Initialize a replay buffer $\mathcal{U} = \emptyset$
5:    // Rollout phase
6:    **for** $t = 0, 1, 2, \ldots$ **do**
7:      Sample output from VLM: $y_t \sim \pi_\theta(\cdot|s_t)$
8:      Parse action: $a_t = f^{\text{parse}}(y_t)$
9:      $r_t, s_{t+1} = $ env.step($a_t$)
10:     $\mathcal{U} = \mathcal{U} \cup \{(s_t, a_t, r_t, y_t)\}$
11:    **end for**
12:    // Counterfactual reasoning phase
13:    Generate nullified sequences: $\{(y^{-i} \cup y^i_{\text{null}})\}_{i=1}^n$
14:    Compute causal weights $\mathcal{B}_{y \to a}$ via Equation (6)
15:    Compute the causal-weighted entropy $\mathcal{H}^{\mathcal{B}}(\pi_\theta(\cdot|s))$
16:    // Model updates phase
17:    Update $\phi$ via CrossEntropy($a, \mathbb{P}_\phi(\cdot|y, \epsilon_a)$)
18:    Update $\theta$ via Equation (7)
19: **end for**
___

While introducing causal-aware adjustments $\mathcal{H}^{\mathcal{B}}(\pi(\cdot|s))$ to guide exploration, CoSo retains the fundamental properties of standard soft RL (Haarnoja et al., 2018), including convergence (Lemma 4.2, Proposition 4.4) and monotonic improvement (Lemma 4.3, Proposition 4.4). Below, we present key theoretical results.

**Lemma 4.2** (Policy Evaluation). *By Equation (7), the Bellman backup operator of CoSo is $\mathcal{T}^{\mathcal{B}}Q(s_t, a_t) = r(s_t, a_t) + \gamma\mathbb{E}_{s_{t+1}\sim P}\Big[\alpha\mathcal{H}^{\mathcal{B}}(\pi(\cdot|s_{t+1})) + \mathbb{E}_{a_{t+1}\sim\pi}[Q(s_{t+1}, a_{t+1})]\Big]$. Let $Q^0 : \mathcal{S} \times \mathcal{A} \to \mathbb{R}$ be an arbitrary initialization (with $|\mathcal{A}| < \infty$), and define the iterative process $Q^{k+1} = \mathcal{T}^{\mathcal{B}}Q^k$. Then $Q^k$ converges to the fixed Q-value as $k \to \infty$.*

**Lemma 4.3** (Policy Improvement). *Let $\pi \in \Pi$ be any policy and $\tilde{\pi}$ be the solution to the maximization of Equation (7). Then, $Q^{\tilde{\pi}}(s, a) \geq Q^\pi(s, a)$ holds for $\forall(s, a) \in \mathcal{S} \times \mathcal{A}$ and $|\mathcal{A}| < \infty$.*

**Proposition 4.4** (Policy Iteration). *Repeatedly applying the steps of Lemma 4.2 and Lemma 4.3 starting from any $\pi \in \Pi$ obtains a sequence of policies $\pi_1, \pi_2, \ldots$ which converges to $\pi^*$, such that, $Q^{\pi^*}(s, a) \geq Q^\pi(s, a)$ for $\forall(s, a) \in \mathcal{S} \times \mathcal{A}$ and $|\mathcal{A}| < \infty$.*

All proofs are provided in Appendix A. Therefore, by leveraging causal importance, CoSo can achieve a more purposeful exploration strategy, leading to faster and more efficient policy optimization for VLM agents, yet it does so without compromising the stability or robustness of the underlying RL algorithm. Moreover, since Equation (7) defines a highly general entropy-based optimization objective, CoSo can be easily adapted to more specific RL objectives (e.g., PPO), making it as flexible as classical maximum entropy RL.

### 4.4. Implementation

In our implementation, we follow prior works (Ouyang et al., 2022; Bai et al., 2024) and adopt a two-stage offline-to-online fine-tuning process to train VLM agents with CoSo.

**Offline phase.** Before RL, we initialize the VLM agent using SFT on task-specific datasets. This step is critical for equipping the agent with prior knowledge of the task, such as the expected input-output formats and valid utterances $y$ interpretable by the parsing function $f^{\text{parse}}$. By grounding the model in task-relevant knowledge, the offline phase ensures the VLM starts with a good initialization, preparing the agent with foundational skills and structures necessary to interact effectively with the environment later.

**Online phase.** In this step, the VLM agent interacts with the environment and leverages CoSo to refine its policy. Unlike the offline phase, where the model's learning is constrained by static datasets, the online phase allows the agent to dynamically adapt to the environment through real-time feedback. The whole online process is outlined in Figure 1 and Algorithm 1. Specifically, we instantiate SCM using a lightweight BERT-based (Devlin et al., 2019) network $\phi$ with 0.01B parameters, which is used to approximate the probabilistic function $\mathbb{P}_\phi(a|y, \epsilon_a)$. Each iteration starts with a rollout phase to sample online data, where the agent interacts with the environment to explore diverse states and transitions. Next, the causal weights $\mathcal{B}_{y \to a}$ are computed via counterfactual reasoning, a critical step in guiding the agent's exploration. This enables the agent to perform more targeted and efficient exploration, focusing on causally significant regions of the policy space rather than random trial-and-error. Finally, we update SCM using cross-entropy loss to improve its approximation of the causal relationships between $y$ and $a$ and update VLM using CoSo objective to enhance the agent's decision-making and exploration.

## 5. Experiments

In this section, we present the empirical evaluations of the proposed CoSo across a range of VLM agent tasks. We choose two existing RL fine-tuning methods, AWR-based DigiRL (Bai et al., 2024) and PPO-based RL4VLM (Zhai et al., 2024), and adapt CoSo directly. Our experiments are designed to demonstrate **(1)** the ability of CoSo to boost the performance of VLM agents over existing RL fine-tuning methods; **(2)** its ability to identify action-critical tokens and achieve more efficient exploration; **(3)** a comprehensive ablation study to understand the contribution of CoSo's components.

### 5.1. Performance Boosting

In this part, we evaluate CoSo's ability to boost the performance of VLM agents over existing RL fine-tuning methods

*Table 1.* Average success rates (%) on AitW General and AitW WebShopping. Results are averaged over 3 random seeds.

| | | AitW General | | AitW WebShopping | | |
| --- | --- | --- | --- | --- | --- | --- |
| | | Train | Test | Train | Test | Avg. |
| **Prompting** | Gemini 1.5 Pro | 32.3 | 16.7 | 6.3 | 11.5 | 16.7 |
| | GPT-4V | 5.2 | 13.5 | 3.1 | 8.3 | 7.5 |
| | Gemini 1.5 Pro + AppAgent | 14.6 | 16.7 | 5.2 | 8.3 | 11.2 |
| | GPT-4V + AppAgent | 13.5 | 17.7 | 12.5 | 8.3 | 13.0 |
| **Learning** | CogAgent | 25.0 | 25.0 | 31.3 | 38.5 | 30.0 |
| | AutoUI + SFT | 38.8 | 37.6 | 40.3 | 41.5 | 39.6 |
| | Online Filtered BC | $53.2 \pm 7.6$ | $50.9 \pm 8.2$ | $59.5 \pm 7.7$ | $52.8 \pm 6.8$ | 54.1 |
| | DigiRL | $64.6 \pm 7.8$ | $62.7 \pm 5.3$ | $68.1 \pm 9.5$ | $64.2 \pm 6.1$ | 64.9 |
| | CoSo (Ours) | $\mathbf{72.9 \pm 5.9}$ | $\mathbf{71.3 \pm 6.5}$ | $\mathbf{77.0 \pm 7.8}$ | $\mathbf{70.5 \pm 8.2}$ | **72.9** |

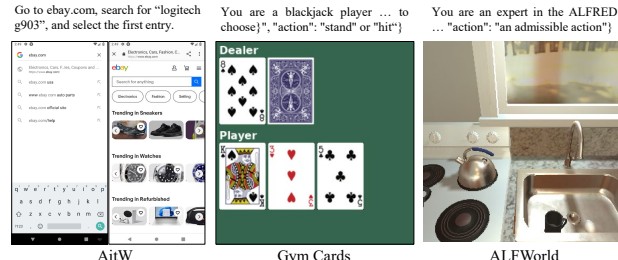

*Figure 2.* Examples of tasks in different environments: (1) *AitW*: mobile control tasks that involve interacting with GUIs to complete multi-step instructions. (2) *Gym Cards*: virtual card-based reasoning tasks requiring numerical understanding, arithmetic operations, and decision-making. (3) *ALFWorld*: embodied AI tasks that combine visual understanding and textual reasoning to perform complex, goal-directed actions in interactive environments.

in both physical and virtual environments.

**Android-in-the-Wild.** We first implement CoSo based on DigiRL (Bai et al., 2024) on top of the AutoUI (Zhang & Zhang, 2024), which is used for fine-tuning mobile agents. We evaluate CoSo on the *General* and *WebShopping* tasks from the Android-in-the-Wild (AitW) (Rawles et al., 2024), which features a rich variety of real-world mobile control tasks that requires multi-step reasoning and interaction with graphical user interfaces (GUIs) on Android emulators, as illustrated in Figure 2.

It can be seen in Table 5 that prompting-based methods, such as Gemini 1.5 Pro (Team et al., 2023), GPT-4V (Achiam et al., 2023), and AppAgent (Zhang et al., 2023), face significant challenges in completing user instructions as they rely on pre-trained knowledge, which constrains their ability to adapt to task-specific requirements. Online fine-tuning approaches, like Online Filtered BC and DigiRL (Bai et al., 2024), outperform offline learning techniques such as CogAgent (Hong et al., 2024) and AutoUI + SFT by facilitating

interactions with Android emulators. This suggests the importance of online adjustment in enhancing VLM agents' ability to adapt to dynamic environments. However, their exploration of all tokens of utterance remains highly inefficient. In contrast, CoSo excels in online fine-tuning, achieving both higher exploration efficiency and performance gains over prior methods. Notably, CoSo achieves an average success rate of 72.9%, an improvement of 12.3% over the previous state-of-the-art.

**Gym Cards.** Next, we implement our approach on RL4VLM (Zhai et al., 2024) on top of the LLaVA-1.6 (Liu et al., 2024b), and evaluate it within the Gym Cards environment (Brockman, 2016), which requires numerical reasoning and arithmetic operations. It includes four tasks: *NumberLine*, involving aligning a value with a specified target using arithmetic operations, *EZPoints* and *Points24* that require constructing arithmetic expressions to achieve target values, and *Blackjack* that introduces uncertainty by requiring decisions based on game states to maximize success. The results are given in Table 2.

Prompting-based methods like Gemini-1.5 Pro and GPT-4V perform comparatively better in Gym Cards than in AitW. This is likely due to the proficiency of VLMs in reasoning tasks related to code and mathematics. Nevertheless, their performance remains inferior to learning-based algorithms like LLaVA-1.6 + SFT (Liu et al., 2024b), CNN + RL, and RL4VLM (Zhai et al., 2024), suggesting a limited scope without fine-tuning. With more purposeful online exploration, CoSo exhibits a greater ability to adapt efficiently to numerical reasoning and arithmetic operation and demonstrates a 9.3% improvement over RL4VLM.

**ALFWorld.** We next test CoSo on the ALFWorld (Shridhar et al., 2021), an embodied environment for testing decision-making in scenarios requiring visual-semantic understanding. The ALFWorld domain includes tasks like *Pick&Place*, *ExamineInLight*, *Clean&Place*, *Heat&Place*, *Cool&Place*,

*Table 2.* Average success rates (%) on Gym Cards and ALFWorld. For Gym Cards, the average performance is computed across all tasks, with each task weighted equally. For ALFWorld, the average success rate is calculated as a weighted mean to account for the varying occurrence probabilities of subtasks in the environment. Results are averaged over 3 random seeds.

| | | **Gym Cards** | | | | | **ALFWorld** | | | | | | |
| | | NL | EZP | P24 | BJ | Avg. | Pick | Look | Clean | Heat | Cool | Pick2 | Avg. |
|---|---|---|---|---|---|---|---|---|---|---|---|---|---|
| **Prompting** | Gemini-1.5 Pro | 82.5 | 2.0 | 0.0 | 30.0 | 28.6 | 34.6 | **16.7** | 0.0 | 0.0 | 0.0 | 12.0 | 13.5 |
| | GPT-4V | 65.5 | 10.5 | 0.0 | 25.5 | 25.4 | 38.2 | 12.1 | 18.8 | 6.7 | 17.8 | 14.6 | 19.4 |
| | LLaVA-1.6 | 14.3 | 2.2 | 0.0 | 21.5 | 9.5 | 16.7 | 0.0 | 0.0 | 0.0 | 0.0 | 0.0 | 5.0 |
| **Learning** | CNN + RL | 87.1 | 0.0 | 0.0 | 38.8 | 31.5 | 0.0 | 0.0 | 0.0 | 0.0 | 0.0 | 0.0 | 0.0 |
| | LLaVA-1.6 + SFT | 24.8 | 23.0 | 2.6 | 23.1 | 18.4 | 39.2 | 0.0 | 14.4 | 11.1 | 0.0 | **28.6** | 17.7 |
| | RL4VLM | 88.4 | **50.0** | 2.5 | 39.3 | 45.1 | **44.7** | 13.8 | 18.3 | **15.8** | 14.3 | 16.8 | 22.7 |
| | CoSo (Ours) | **100.0** | **50.0** | **5.8** | **41.5** | **49.3** | 42.5 | **16.7** | 21.4 | 11.9 | **22.1** | 25.9 | **26.5** |

and *PickTwo&Place*, which involve reasoning over high-level goals and executing step-by-step plans in visual environments. As depicted in Figure 2, tasks such as Pick&Place require finding an object (e.g., a kettle) and placing it in a specified location (e.g., a countertop), involving navigation and object manipulation.

The results in Table 2 demonstrate that online fine-tuning in RL4VLM plays a crucial role in task completion, allowing the agent to dynamically adjust its strategies during training. It achieves the state-of-the-art result (22.7%), surpassing prompting-based methods like Gemini-1.5 Pro and GPT-4V. Consistent with the results observed in AitW and Gym Cards, CoSo leverages causal-aware exploration to attain the best success rate of 26.5%, an improvement of 16.7%. These findings highlight the effectiveness of CoSo in fine-tuning VLM agents across various task domains.

## 5.2. Efficient Exploration

We now evaluate CoSo's ability to identify action-critical tokens and facilitate more efficient exploration. In this experiment, we first analyze the counterfactual results of CoSo and present the AitW results in Figure 3 (above). See Appendix E for additional results in Gym Cards and ALFWorld. As observed, CoSo effectively identifies the causal importance of individual tokens, highlighting in red those that truly shape the action. Notably, these key tokens constitute only a small proportion of the total (less than 10%), whereas over 80% of the tokens have weights below 0.2, indicating minimal influence on the final action. Prioritizing exploration of high-weight tokens proves beneficial, as variations in these tokens are more likely to drive meaningful action changes, thereby greatly improving exploration efficiency.

To further validate this, we test the VLM agent in a challenging Android mobile control scenario that demands efficient exploration to handle errors and recover from mistakes. As illustrated in Figure 3 (below), the agent mistakenly clicks the "Share" page upon entering the Walmart homepage. Ef-

fective recovery in this situation requires corrective actions, such as pressing the "Back" or "Home" button. In this scenario, we compare CoSo's action exploration performance against other algorithms, including naive RL and RL with entropy regularization. Each algorithm samples multiple utterances to assess whether the agent can produce actions that enable recovery from its own mistake.

As demonstrated in Figure 3 (below), naive RL, without entropy-based exploration, tends to repeatedly select the "DUAL POINT" (i.e., click on the "Search Walmart") even when this button is unclickable in the given situation, results in repetitive and ineffective exploration. Indeed, naive RL can sample different utterances with varying planning results and coordinates. However, as the exploration space of complete utterances is vast, most token changes cannot result in meaningful variations in actions. Introducing entropy regularization into RL encourages more diverse action sampling and facilitates the generation of varied action plans. However, since it uniformly assigns uncertainty to all tokens during optimization, the agent often explores irrelevant tokens or produces outputs in incorrect formats, as highlighted by the red examples in Figure 3 (below). In contrast, CoSo effectively identifies the causal importance of specific tokens, such as "DUAL" in Figure 3 (above). Hence, CoSo can selectively introduce uncertainty to action-critical tokens by weighting their entropy during optimization (Equation (7)). This ensures more efficient sampling and aligns the sampling of utterances more closely with the sampling of final actions. As demonstrated, CoSo successfully samples one of the correct recovery actions ("Home"), which greatly contributes to the effectiveness of RL online fine-tuning. See more exploration comparison in Appendix F.

## 5.3. Ablation Study

Next, we conducted an ablation study on the WebShopping and NumberLine tasks, comparing standard RL (RL), RL with naive entropy (RL + $\mathcal{H}$), and CoSo (RL + $\mathcal{H}^{\mathcal{B}}$) to

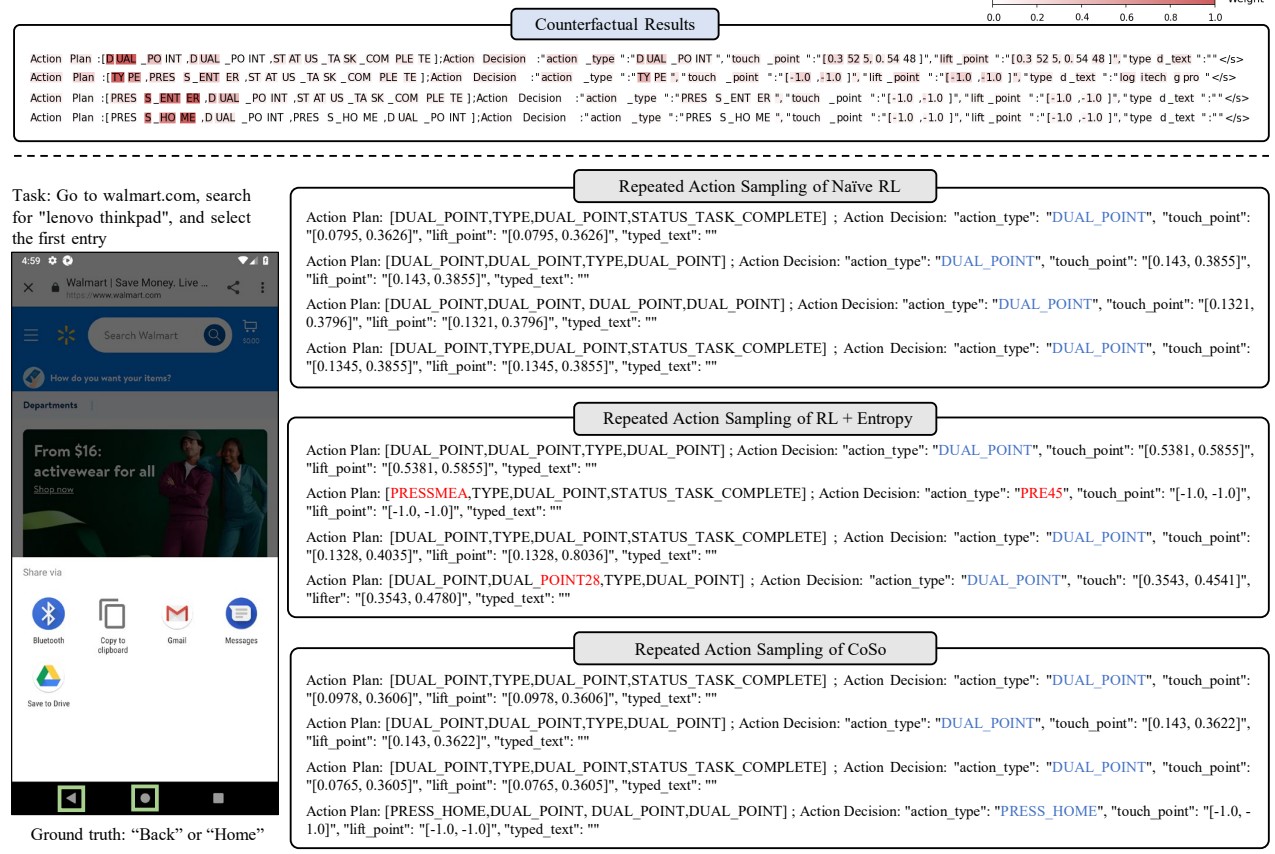

Figure 3. Counterfactual results (above) and repeated action sampling (below). For the counterfactual results, causal weights $\mathcal{B}_{y \to a}$ are normalized to [0, 1]. In the repeated action sampling, the action fields are highlighted in blue, while format errors are indicated in red.

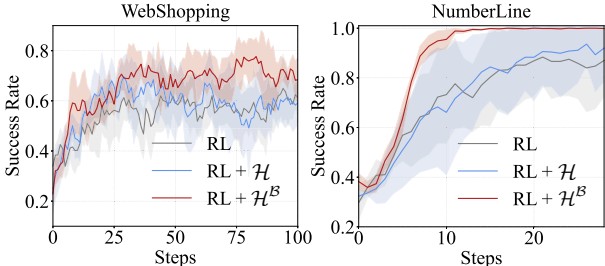

Figure 4. Ablation results. The curve of WebShopping is smoothed with exponential weighting over the x-axis.

evaluate the impact of each component on performance. The results are presented in Figure 4.

As shown, RL with naive entropy provided only marginal improvement over standard RL. This is because applying uniform randomness to all tokens fails to focus on those that are truly critical for action generation. It also leads to inefficient exploration and increases the likelihood of generating invalid or irrelevant outputs, as discussed in Figure 3. As a result, naive entropy struggles to address the exploration issue posed by the textual action space of VLM agents. After introducing causal weights, CoSo introduces meaningful variability where it matters most, which can drastically reduce the exploration space and make the process more efficient. As shown in Figure 4, CoSo outperforms standard RL and RL with naive entropy in terms of both convergence speed and final task success.

## 6. Conclusions and Limitations

In this paper, we address the challenge of inefficient exploration in RL online fine-tuning of VLM agents. We proposed a novel and effective solution, CoSo, which leverages counterfactual reasoning to assess the causal importance of tokens. By prioritizing the exploration of action-critical tokens instead of random trial and error, CoSo introduces a principled approach to efficient fine-tuning. Its general objective enables adaptability to various RL objectives, making it a scalable framework for future advancements in online RL fine-tuning. Theoretical analysis and empirical results both support the effectiveness of CoSo. Notably, it consistently outperforms existing RL fine-tuning methods across diverse agent tasks, including Android device control, card

gaming, and embodied AI. Despite its strong performance, the longest utterances in our experiments contain less than 300 tokens. Its effectiveness on ultra-long CoT has not yet been explored. Investigating strategies to extend CoSo to such settings, possibly through hierarchical reasoning, is a promising direction for future work.

## Acknowledgements

This research is supported by the National Research Foundation, Singapore under its Industry Alignment Fund – Prepositioning (IAF-PP) Funding Initiative. Any opinions, findings and conclusions or recommendations expressed in this material are those of the author(s) and do not reflect the views of National Research Foundation, Singapore. This research is supported by the RIE2025 Industry Alignment Fund – Industry Collaboration Projects (IAF-ICP) (Award I2301E0026), administered by A*STAR, as well as supported by Alibaba Group and NTU Singapore through Alibaba-NTU Global e-Sustainability CorpLab (ANGEL).

## Impact Statement

This paper presents work whose goal is to advance the field of Machine Learning. There are many potential societal consequences of our work, none which we feel must be specifically highlighted here.

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

# A. Proofs

## A.1. Equation (2)

*Proof.* The entropy $\mathcal{H}(\pi(\cdot|\boldsymbol{s}))$ over action is equivalent to the entropy of the output utterance $\boldsymbol{y} = (y^1, y^2, \ldots, y^n)$, which is defined as:

$$\mathcal{H}(\boldsymbol{y}) = -\mathbb{E}_{\boldsymbol{y}}\left[\log \pi_\theta(\boldsymbol{y}|\boldsymbol{s})\right] \tag{A.1}$$

$$= -\mathbb{E}_{\boldsymbol{y}}\left[\sum_{i=1}^{n} \log \pi_\theta(y^i|\boldsymbol{y}^{1:i-1}, \boldsymbol{s})\right] \tag{A.2}$$

$$= \sum_{i=1}^{n} -\mathbb{E}_{\boldsymbol{y}}\left[\log \pi_\theta(y^i|\boldsymbol{y}^{1:i-1}, \boldsymbol{s})\right] \tag{A.3}$$

$$= \sum_{i=1}^{n} -\mathbb{E}_{\boldsymbol{y}^{1:i}}\left[\log \pi_\theta(y^i|\boldsymbol{y}^{1:i-1}, \boldsymbol{s})\right] \tag{A.4}$$

$$= \sum_{i=1}^{n}\left[-\sum_{\boldsymbol{y}^{1:i}}\left[\pi_\theta(\boldsymbol{y}^{1:i}|\boldsymbol{s}) \log \pi_\theta(y^i|\boldsymbol{y}^{1:i-1}, \boldsymbol{s})\right]\right] \tag{A.5}$$

$$= \sum_{i=1}^{n} \mathcal{H}(y^i|\boldsymbol{y}^{1:i-1}), \tag{A.6}$$

where $\mathcal{H}(y^i|\boldsymbol{y}^{1:i-1})$ denotes the conditional entropy of $y^i$ given $\boldsymbol{y}^{1:i-1}$. $\qquad\square$

## A.2. Lemma 4.2

**Lemma 4.2** (Policy Evaluation). *Derived from Equation (7), the Bellman backup operator of CoSo is given by* $\mathcal{T}^{\mathcal{B}}Q(\boldsymbol{s}_t, \boldsymbol{a}_t) = r(\boldsymbol{s}_t, \boldsymbol{a}_t) + \gamma\mathbb{E}_{\boldsymbol{s}_{t+1}\sim P}\left[\alpha\mathcal{H}^{\mathcal{B}}(\pi(\cdot|\boldsymbol{s}_{t+1})) + \mathbb{E}_{\boldsymbol{a}_{t+1}\sim\pi}[Q(\boldsymbol{s}_{t+1}, \boldsymbol{a}_{t+1})]\right].$ *Let* $Q^0 : \mathcal{S} \times \mathcal{A} \to \mathbb{R}$ *be an arbitrary initialization (with* $|\mathcal{A}| < \infty$*), and define the iterative process* $Q^{k+1} = \mathcal{T}^{\mathcal{B}}Q^k$. *Then* $Q^k$ *converges to the fixed Q-value as* $k \to \infty$.

*Proof.* The $Q$-function of Equation (7) is given by

$$Q(\boldsymbol{s}, \boldsymbol{a}) = \mathbb{E}_{(\boldsymbol{s}_1, \boldsymbol{a}_1, \ldots)\sim\rho^\pi}\left[r(\boldsymbol{s}_0, \boldsymbol{a}_0) + \sum_{t=1,2,\ldots} \gamma^t\left(r(\boldsymbol{s}_t, \boldsymbol{a}_t) + \alpha\mathcal{H}^{\mathcal{B}}(\pi(\cdot|\boldsymbol{s}_t))\right)\Big|\boldsymbol{s}_0 = \boldsymbol{s}, \boldsymbol{a}_0 = \boldsymbol{a}\right] \tag{A.7}$$

$$= \mathbb{E}_{(\boldsymbol{s}_1, \boldsymbol{a}_1, \ldots)\sim\rho^\pi}\left[r(\boldsymbol{s}_0, \boldsymbol{a}_0) + \gamma\Big(r(\boldsymbol{s}_1, \boldsymbol{a}_1) + \alpha\mathcal{H}^{\mathcal{B}}(\pi(\cdot \mid \boldsymbol{s}_1)) + \gamma\Big(r(\boldsymbol{s}_2, \boldsymbol{a}_2) + \ldots\Big)\Big)\Big|\boldsymbol{s}_0 = \boldsymbol{s}, \boldsymbol{a}_0 = \boldsymbol{a}\right].$$

This recursive structure simplifies to:

$$Q(\boldsymbol{s}, \boldsymbol{a}) = r(\boldsymbol{s}, \boldsymbol{a}) + \gamma\mathbb{E}_{\boldsymbol{s}'\sim P}\left[\alpha\mathcal{H}^{\mathcal{B}}(\pi(\cdot \mid \boldsymbol{s}')) + \mathbb{E}_{\boldsymbol{a}'\sim\pi}[Q(\boldsymbol{s}', \boldsymbol{a}')]\right]. \tag{A.8}$$

We see that the $Q$-function can be updated iteratively using the operator:

$$\mathcal{T}^{\mathcal{B}}Q(\boldsymbol{s}_t, \boldsymbol{a}_t) = r(\boldsymbol{s}_t, \boldsymbol{a}_t) + \gamma\mathbb{E}_{\boldsymbol{s}_{t+1}\sim P}\left[\alpha\mathcal{H}^{\mathcal{B}}(\pi(\cdot|\boldsymbol{s}_{t+1})) + \mathbb{E}_{\boldsymbol{a}_{t+1}\sim\pi}[Q(\boldsymbol{s}_{t+1}, \boldsymbol{a}_{t+1})]\right]. \tag{A.9}$$

Then, we re-organize the operator:

$$\mathcal{T}^{\mathcal{B}}Q(\boldsymbol{s}_t, \boldsymbol{a}_t) = r(\boldsymbol{s}_t, \boldsymbol{a}_t) + \gamma\mathbb{E}_{\boldsymbol{s}_{t+1}\sim P}\left[\alpha\mathcal{H}^{\mathcal{B}}(\pi(\cdot|\boldsymbol{s}_{t+1})) + \mathbb{E}_{\boldsymbol{a}_{t+1}\sim\pi}[Q(\boldsymbol{s}_{t+1}, \boldsymbol{a}_{t+1})]\right] \tag{A.10}$$

$$= r(\boldsymbol{s}_t, \boldsymbol{a}_t) + \gamma\mathbb{E}_{\boldsymbol{s}_{t+1}\sim P}\left[\alpha\mathcal{H}^{\mathcal{B}}(\pi(\cdot|\boldsymbol{s}_{t+1}))\right] + \mathbb{E}_{\boldsymbol{a}_{t+1}\sim\pi}\left[Q(\boldsymbol{s}_{t+1}, \boldsymbol{a}_{t+1})\right] \tag{A.11}$$

$$= r^{\mathcal{B}}(\boldsymbol{s}_t, \boldsymbol{a}_t) + \gamma\mathbb{E}_{\boldsymbol{a}_{t+1}\sim\pi}\left[Q(\boldsymbol{s}_{t+1}, \boldsymbol{a}_{t+1})\right], \tag{A.12}$$

where $r^{\mathcal{B}}(\boldsymbol{s}_t, \boldsymbol{a}_t)$ is the CoSo-augmented reward. The key observation is that $\mathcal{H}^{\mathcal{B}}(\pi(\cdot \mid \boldsymbol{s}))$ is bounded above for any policy $\pi$ and state $\boldsymbol{s}$ since it is a weighted sum of the (per-token) discrete entropies $\mathcal{H}(y^i|\boldsymbol{y}^{1:i-1})$. Specifically, for each token $y^i$ and weights $\mathcal{B}^i_{\boldsymbol{y} \to \boldsymbol{a}} \geq 0$, we have

$$0 \leq \sum_{i=1}^n \mathcal{B}^i_{\boldsymbol{y} \to \boldsymbol{a}} \cdot \mathcal{H}(y^i|\boldsymbol{y}^{1:i-1}) \leq \left(\max_i \mathcal{B}^i_{\boldsymbol{y} \to \boldsymbol{a}}\right) \cdot n \log |\mathcal{V}|, \tag{A.13}$$

where $\mathcal{V}$ is the token (vocabulary) set. As such, $r^{\mathcal{B}}(\boldsymbol{s}_t, \boldsymbol{a}_t)$ is bounded by $\left[r_{\min}, r_{\max} + \alpha \left(\max_i \mathcal{B}^i_{\boldsymbol{y} \to \boldsymbol{a}}\right) \cdot n \log |\mathcal{V}|\right]$ which satisfies the boundedness requirement of standard policy evaluation (Sutton & Barto, 2018). As a consequence, $\mathcal{T}^{\mathcal{B}}$ remains a contraction mapping in an appropriate norm (e.g., an $\ell_\infty$ norm), guaranteeing convergence of $Q^k$. $\qquad \square$

## A.3. Lemma 4.3

**Lemma 4.3** (Policy Improvement). *Let $\pi \in \Pi$ be any policy and $\tilde{\pi}$ be the solution to the maximization of Equation (7). Then, $Q^{\tilde{\pi}}(\boldsymbol{s}, \boldsymbol{a}) \geq Q^\pi(\boldsymbol{s}, \boldsymbol{a})$ holds for $\forall(\boldsymbol{s}, \boldsymbol{a}) \in \mathcal{S} \times \mathcal{A}$ and $|\mathcal{A}| < \infty$.*

*Proof.* By Equation (A.7), the CoSo policy objective can be defined as

$$\max_\pi \sum_t \mathbb{E}_{\boldsymbol{a}_t \sim \pi} \left[Q^\pi(\boldsymbol{s}_t, \boldsymbol{a}_t) + \alpha \mathcal{H}^{\mathcal{B}}(\pi(\cdot|\boldsymbol{s}_t))\right] \tag{A.14}$$

If consider greedy maximization from $\pi$ to $\tilde{\pi}$, we have

$$\mathbb{E}_{\boldsymbol{a} \sim \tilde{\pi}} \left[Q^\pi(\boldsymbol{s}, \boldsymbol{a})\right] + \alpha \mathcal{H}^{\mathcal{B}}(\tilde{\pi}(\cdot|\boldsymbol{s})) \geq \mathbb{E}_{\boldsymbol{a} \sim \pi} \left[Q^\pi(\boldsymbol{s}, \boldsymbol{a})\right] + \alpha \mathcal{H}^{\mathcal{B}}(\pi(\cdot|\boldsymbol{s})). \tag{A.15}$$

In a similar way to the proof of the soft policy improvement (Haarnoja et al., 2017), we derive the following inequality (we denote $r(\boldsymbol{s}_t, \boldsymbol{a}_t)$ as $r_t$ for convenience):

$$
\begin{aligned}
Q^\pi(\boldsymbol{s}, \boldsymbol{a}) &= r_0 + \gamma \mathbb{E}_{\boldsymbol{s}_1} \left[\alpha \mathcal{H}^{\mathcal{B}}(\pi(\cdot|\boldsymbol{s}_1)) + \mathbb{E}_{\boldsymbol{a}_1 \sim \pi}[Q^\pi(\boldsymbol{s}_1, \boldsymbol{a}_1)]\right] \\
&\leq \mathbb{E}_{\boldsymbol{s}_1} \left[r_0 + \gamma \left(\alpha \mathcal{H}^{\mathcal{B}}(\tilde{\pi}(\cdot \mid \boldsymbol{s}_1)) + \mathbb{E}_{\boldsymbol{a}_1 \sim \tilde{\pi}}[Q^\pi(\boldsymbol{s}_1, \boldsymbol{a}_1)]\right)\right] \\
&\leq \mathbb{E}_{\boldsymbol{s}_1, \boldsymbol{a}_1 \sim \tilde{\pi}} \left[r_0 + \gamma \left(\alpha \mathcal{H}^{\mathcal{B}}(\tilde{\pi}(\cdot \mid \boldsymbol{s}_1)) + r_1 + \gamma^2 \mathbb{E}_{\boldsymbol{s}_2} \left[\alpha \mathcal{H}^{\mathcal{B}}(\tilde{\pi}(\cdot \mid \boldsymbol{s}_2)) + \mathbb{E}_{\boldsymbol{a}_2 \sim \tilde{\pi}}[Q^\pi(\boldsymbol{s}_2, \boldsymbol{a}_2)]\right]\right)\right] \\
&\vdots \\
&\leq \mathbb{E}_{\boldsymbol{s}_1, \boldsymbol{a}_1 \sim \tilde{\pi}, \boldsymbol{s}_2, \boldsymbol{a}_2 \sim \tilde{\pi}, \dots} \left[r_0 + \sum_{t=1,2,\dots} \gamma^t \left(r_t + \alpha \mathcal{H}^{\mathcal{B}}(\tilde{\pi}(\cdot \mid \boldsymbol{s}_t))\right)\right] \\
&= Q^{\tilde{\pi}}(\boldsymbol{s}, \boldsymbol{a}).
\end{aligned}
$$

$\qquad \square$

## A.4. Proposition 4.4

**Proposition 4.4** (Policy Iteration). *Repeatedly applying the steps of Lemma 4.2 and Lemma 4.3 starting from any $\pi \in \Pi$ obtains a sequence of policies $\pi_1, \pi_2, \dots$ which converges to $\pi^*$, such that, $Q^{\pi^*}(\boldsymbol{s}, \boldsymbol{a}) \geq Q^\pi(\boldsymbol{s}, \boldsymbol{a})$ for $\forall(\boldsymbol{s}, \boldsymbol{a}) \in \mathcal{S} \times \mathcal{A}$ and $|\mathcal{A}| < \infty$.*

*Proof.* By Lemma 4.3, the sequence of $Q$-function $(Q^{\pi_1}, Q^{\pi_2}, \dots)$ is monotonically improving. Moreover, it converges since the sequence of $Q$-function is bounded from above as discussed in Lemma 4.2. Let $Q^*$ be its limit. By continuity of the contraction operator (Lemma 4.2) and the construction of each improved policy (Lemma 4.3), the limiting $Q^*$ must be the unique $Q^{\pi^*}$ satisfying $Q^{\pi^*}(\boldsymbol{s}, \boldsymbol{a}) \geq Q^{\pi_k}(\boldsymbol{s}, \boldsymbol{a}), \forall k$ and $\forall(\boldsymbol{s}, \boldsymbol{a}) \in \mathcal{S} \times \mathcal{A}$. Consequently, $\pi^*$ is a fixed-point policy that cannot be further improved. $\qquad \square$

# B. Experiment Details

## B.1. AitW

**Overview.** AitW provides a diverse set of real-world mobile control tasks designed to evaluate the ability to perform complex reasoning and interact with GUIs. We choose two sets in our experiments: *General* and *WebShopping*. The General set focuses on information search and basic application usage, such as searching for news in Chile and opening Gmail. The training set includes all 545 tasks, while the test set comprises the first 96 tasks. The maximum allowed steps per task is 10. The WebShopping set comprises search instructions on various shopping websites, such as "go to ebay.com, search for "logitech g903", and select the first entry". The training set consists of 438 tasks, with 96 tasks reserved for testing. The maximum allowed steps per task is 20.

**Observations.** The observation space consists of the Android GUI screenshot and textual task descriptions provided as user instructions, and action history.

**Actions.** The action space includes various types of mobile operations. `Dual Point`: represents both click and slide actions through touch and lift points, which uses normalized coordinates to ensure adaptability across different screen resolutions and device sizes. Coordinates are scaled to the range $[0, 1]$ relative to the screen dimensions. `Type`: allows text input; `Home`, `Back`, `Enter`: standard navigation actions.

**Baselines.** We compare the performance of several baseline models using scores reported in Table 1 of (Bai et al., 2024). The baselines include: Gemini 1.5 Pro (with and without AppAgent), GPT-4V (with and without AppAgent), CogAgent. We run AutoUI + SFT, Online Filtered BC, and DigiRL, an online RL fine-tuning method based on AWR (Peng et al., 2019), using the official implementation[1].

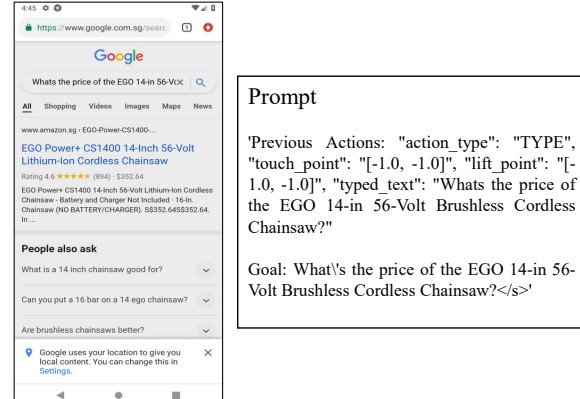

*Figure B.1.* Example of observation and prompt in AitW.

**CoSo.** We implement CoSo on DigiRL (Bai et al., 2024) with a 0.01B BERT-based SCM. Similar to the VLM agent, SCM is trained in two stages: offline and online. The learning rate for SCM is set to $1 \times 10^{-5}$, and the entropy coefficient $\alpha$ is set to 1.0. All other hyperparameters remain consistent with the DigiRL setup.

## B.2. Gym Cards

**Overview.** Gym Cards is a "gym-like" environment designed to evaluate a VLM's vision recognition and arithmetic reasoning. It consists of four tasks, including *NumberLine*, *EZPoints*, *Points24*, and *Blackjack*, designed to evaluate the decision-making capabilities of VLM agents. The first three tasks are deterministic, focusing on number recognition and arithmetic reasoning with increasing complexity. Blackjack, on the other hand, is a stochastic task that requires reasoning based on visual inputs and adapting to uncertain outcomes.

**Observations.** The observation space includes visual inputs, such as images of cards or numbers, and textual task descriptions provided as prompts. See an example of the observation and prompt in Figure B.2.

**Actions.** The action space varies by task. In the NumberLine, actions are + and −, representing increments or decrements to the current number. In the EZPoints and Points24, actions include natural numbers (`1-10`) and arithmetic operators (+, *, =, etc.). In the Blackjack task, actions are `stand` (to hold) and `hit` (to request a new card).

**Baselines.** The baseline models evaluated in our experiments include Gemini 1.5 Pro, GPT4-V, CNN + RL, LLaVA-1.6 + SFT (results sourced from Table 2 in (Zhai et al., 2024)). Additionally, we run LLaVA-1.6 and RL4VLM, an online RL fine-tuning method based on PPO (Schulman et al., 2017), using the the official implementation[2] of RL4VLM.

**CoSo.** We implement CoSo on RL4VLM (Zhai et al., 2024) using a 0.01B BERT-based SCM. Similar to AitW, the learning

---

[1] https://github.com/DigiRL-agent/digirl
[2] https://github.com/RL4VLM/RL4VLM

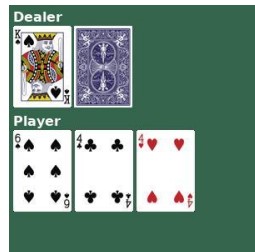

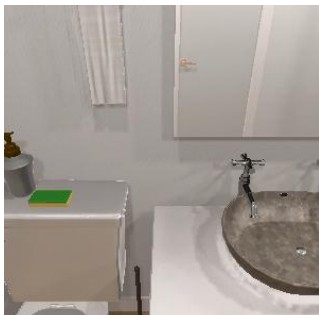

**Prompt**

You are a blackjack player. Given the current state, you can choose between ['stand', 'hit']. Your response should be a valid json file in the following format:

{\"thoughts\": {first describe your total points and the dealer's total points then think about which action to choose},

\"action\": \"stand\" or \"hit\"

*Figure B.2.* Example of observation and prompt in Gym Cards.

rate for SCM was set to $1 \times 10^{-5}$, with the entropy coefficient $\alpha$ set to 1.0. All other hyperparameters, including the configuration of PPO and the scaling factor $\lambda$ for CoT reasoning, were consistent with the RL4VLM configuration.

### B.3. ALFWorld

**Overview.** ALFWorld is an embodied AI environment designed to evaluate decision-making tasks requiring visual semantic reasoning. The environment integrates text-based interaction with visual observations, allowing agents to complete household-related tasks in a simulated environment. There are 4639 games categorized into six types: *Pick & Place*, *Examine in Light*, *Clean & Place*, *Heat & Place*, *Cool & Place*, and *Pick Two & Place*, which involve interacting with objects and their surroundings in contextually appropriate ways.

**Observations.** At each state $s_t$ in ALFWorld, the agent observes an RGB image and a text-based description of the environment. These observations provide the necessary visual and semantic information for the agent to make decisions. The text-based descriptions are inherited from the Text World environment, offering detailed contextual information about the objects and receptacles in the scene. See an example of the observation and prompt in Figure B.3.

**Prompt**

Your are an expert in the ALFRED Embodied Environment. You are also given the following text description of the current scene: ['-= Welcome to TextWorld, ALFRED! =-\\n\\nYou are in the middle of a room. Looking quickly around you, you see a countertop 1, a sink 1a sinkbasin 1, a drawer 1, a drawer 2, a drawer 3, a drawer 4, a drawer 5, a drawer 6, a sink 2, a sinkbasin 2, a drawer 7, a drawer 8, and a toilet 1.\\n\\nYour task is to: find two toiletpaper and put them in drawer']. Your task is to put two toiletpaper in drawer. Your admissible actions of the current situation are: ['go to countertop 1'\n 'go to sink 1'\n 'go to sinkbasin 1'\n 'go to drawer 1'\n 'go to drawer 2'\n 'go to drawer 3'\n 'go to drawer 4'\n 'go to drawer 5'\n 'go to drawer 6'\n 'go to sink 2'\n 'go to sinkbasin 2'\n 'go to drawer 7'\n 'go to drawer 8'\n 'go to toilet 1'\n 'inventory'\n 'look']. Your response should be a valid json file in the following format: \n\\{\"thoughts\": \"{first describe what do you see in the image using the text description, then carefully think about which action to complete the task. }\", \n\"action\": \"{an admissible action}\"\n\\

*Figure B.3.* Example of observation and prompt in ALFWorld.

**Actions.** The action space consists of admissible text-based commands, which vary across different tasks. For examples, the action template includes `"go to {recep}"`, `"open {recep}"`, `"close {recep}"`, `"take {obj} from {recep}"`, `"put {obj} in/on {recep}"`, `"use {lamp}"`, `"heat {obj} with {microwave}"`, `"cool {obj} with {fridge}"`, `"clean {obj} with {cleaner}"`, `"slice {obj} with {knife}"`, `"inventory"`, `"look"`, `"examine {obj}"`, and `"examine {recep}"`. Here, `{obj}` represents objects and `{recep}` represents receptacles. The action space is state-dependent, meaning that only certain actions are available or valid in specific states, depending on the agent's current task and environment configuration.

**Baselines.** The baseline models evaluated include Gemini 1.5 Pro, GPT4-V, CNN + RL, and LLaVA-1.6 + SFT. Results for these baselines are sourced from Table 2 in (Zhai et al., 2024). Additionally, we run LLaVA-1.6 and RL4VLM using the official implementation of RL4VLM.

**CoSo.** We implement CoSo on RL4VLM (Zhai et al., 2024) using a 0.01B BERT-based SCM. The learning rate for SCM was set to $1 \times 10^{-5}$, and the entropy coefficient $\alpha$ was set to 1.0. All other hyperparameters, including the configuration of PPO and the scaling factor $\lambda$ for CoT reasoning, were consistent with the RL4VLM configuration.

## C. Extension to LLM Agents

While our method is primarily designed for VLM agents, it naturally extends to LLM agents operating in purely textual environments. To demonstrate this, we conduct experiments on AlfredTWEnv, a fully text-based environment within the ALFWorld suite (Shridhar et al., 2021).

We implement RL4VLM and CoSo using `Qwen2.5-7B-Instruct` (Yang et al., 2024) and train the LLM agents from scratch (i.e., without SFT) over 12,000 environment steps. Table C.1 summarizes the results, showing that CoSo outperforms the RL baseline in average task success.

*Table C.1.* Performance comparison between RL4VLM and CoSo on the AlfredTWEnv with LLM agents.

| AlfredTWEnv | Pick | Look | Clean | Heat | Cool | Pick2 | Avg. |
|---|---|---|---|---|---|---|---|
| RL4VLM (LLM Agent) | 62.9 | **38.5** | 35.0 | 33.3 | 25.9 | **11.1** | 32.8 |
| CoSo (LLM Agent) | **77.1** | 24.2 | **40.7** | **37.5** | **35.3** | 7.0 | **39.6** |

## D. Computational Budget

In this part, we evaluate the computational budge of CoSo's online fine-tuning and compare its overhead with the baseline RL method (Zhai et al., 2024) and RL with naive entropy regularization on NumberLine. The evaluation includes metrics such as parameter count, memory usage, and training time, with all results obtained using NVIDIA H100 cores.

As summarized in Table D.1, CoSo introduces a modest computational overhead due to the need to compute causal weights for each token in the generated utterances. However, this overhead is relatively small because the SCM is implemented using lightweight BERT-based networks. Specifically, CoSo adds only 0.01 B parameters ($< 0.2\%$), 0.7 GB of GPU memory ($< 2\%$), and 0.5 H100 GPU hours ($< 4\%$). This volume of the budget is modest while achieving significant performance improvements in exploration efficiency and task success rates, as shown in Section 5.1 and Section 5.2.

*Table D.1.* Computational budget including parameter count (B), GPU memory usage (GB), and training time (H100 GPU hours).

| Method | Parameter | | | GPU Memory | | | Training Time | | | |
|---|---|---|---|---|---|---|---|---|---|---|
| | VLM | SCM | Total | VLM | SCM | Total | Rollout | Update | $\mathcal{H}^{\mathcal{B}}$ | Total |
| RL | 7.96 | N/A | 7.96 | 37.0 | N/A | 37.0 | 10.4 | 3.5 | N/A | 13.9 |
| RL + Entropy | 7.96 | N/A | 7.96 | 37.0 | N/A | 37.0 | 10.4 | 3.6 | N/A | 14.0 |
| CoSo | 7.96 | 0.01 | 7.97 | 37.0 | 0.7 | 37.7 | 10.4 | 3.6 | 0.5 | 14.5 |

# E. Visual Counterfactual Results

We present the visual counterfactual results in Figure E.1 and the distribution of weights in Figure E.2. As observed, the SCM can effectively identify the tokens that truly shape the outcome action. These tokens represent a small fraction ($<$10%) of the total, while over 80% of the tokens have weights in the range of [0, 0.2], which indicates minimal impact on the action. Moreover, ALFWorld results indicate that utterances with longer CoT may contain a higher proportion of low-impact tokens. By reducing the exploration of these tokens, CoSo substantially improves action exploration efficiency.

*Figure E.1.* Visual counterfactual results on AitW, Gym Cards, and ALFWorld. Darker colors indicate greater weights.

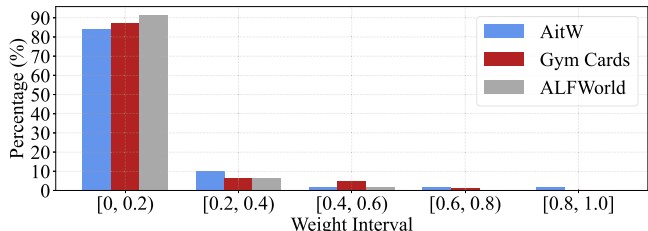

*Figure E.2.* Distribution of counterfactual weights $\mathcal{B}_{y \to a}$ (normalized)

# F. More Exploration Comparison

Ground truth: cup in diningtable 1.

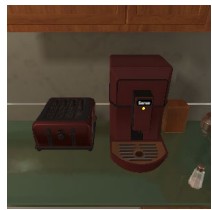

Your are an expert in the ALFRED Embodied Environment. You are also given the following text description of the current scene: [\'You arrive at loc 7. On the coffeemachine 1, you see a saltshaker 1, a spoon 1, and a bread 1.\']. Your task is to put a hot cup in diningtable. Your admissible actions of the current situation are: [\'go to fridge 1\'\n \'go to cabinet 1\'\n \'go to sink 1\'\n \'go to countertop 1\'\n \'go to sinkbasin 1\'\n \'go to cabinet 2\'\n \'go to toaster 1\'\n \'go to cabinet 3\'\n \'go to cabinet 4\'\n \'go to cabinet 5\'\n \'go to drawer 1\'\n \'go to drawer 2\'\n \'go to microwave 1\'\n \'go to stoveburner 1\'\n \'go to cabinet 6\'\n \'go to stoveburner 2\'\n \'go to stoveburner 3\'\n \'go to garbagecan 1\'\n \'go to cabinet 7\'\n \'go to cabinet 8\'\n \'go to stoveburner 4\'\n \'go to diningtable 1\'\n \'take saltshaker 1 from coffeemachine 1\'\n \'take spoon 1 from coffeemachine 1\'\n \'take bread 1 from coffeemachine 1\'\n \'inventory\'\n \'look\'\n \'examine coffeemachine 1\']. Your response should be a valid json file in the following format: \n\{\n"thoughts": "{first describe what do you see in the image using the text description, then carefully think about which action to complete the task. }", \n"action": "{an admissible action}"\n\\}

### Naïve RL

1. '{\n "thoughts": "Based on the text description, I see a saltshaker, a spoon, and a piece of bread on the coffee machine. To put a hot cup on the dining table, I first need to find a cup and heat it. Since there is no mention of a cup on the coffee machine, I should look for a cup in the cabinets or on the countertops. After acquiring a cup, I would need to use the coffee machine to heat the water. However, the first step is to find a cup.",\n "action": "go to cabinet 1"\n}\n'

2. '{\n "thoughts": "Based on the text description, I see a coffee machine with a saltshaker, a spoon, and a piece of bread on it. To put a hot cup on the dining table, I first need to find a cup, fill it with coffee, and then heat it up. Since the coffee machine is already in view, I should look for a cup nearby, possibly in a cabinet or on a countertop. After acquiring a cup, I can then use the coffee machine to heat the water. However, the first step is to find a cup, so I will go to a location where cups are commonly stored, such as a cabinet.",\n "action": "go to cabinet 1"\n}\n'

3. '{\n "thoughts": "In the image, I see a coffee machine with a toaster to the left and a saltshaker, a spoon, and a piece of bread on top of it. To complete the task of putting a hot cup on the dining table, I need to find a cup, fill it with coffee, and then heat it up. Since there is no direct action to \'heat cup\' or \'take cup\', I should look for a cup first, which might be in a cabinet or on a countertop. The action \'go to diningtable 1\' will bring me closer to the dining table, but I need the cup before I can place it there. Therefore, I should look for a cup in the cabinets or on the countertops.",\n "action": "go to cabinet 1"\n}\n'

4. '{\n "thoughts": "Based on the text description, I am at loc 7 and there is a coffee machine with a saltshaker, a spoon, and a piece of bread on it. To put a hot cup on the dining table, I first need to find a cup and heat it. Since there\'s no mention of a cup on the coffee machine, I should look for a cup in the cabinets or on the countertops. After securing a cup, I would need to use the coffee machine to heat the water or the microwave to heat the cup if available. Since \'go to microwave 1\' is an admissible action, it suggests that there is a microwave in the environment. However, I need to find a cup first, so I should go to a location where cups are typically stored, such as a cabinet.",\n "action": "go to cabinet 1"\n}\n'

5. '{\n "thoughts": "I can see a coffee machine and on it next to a toaster, a tolte. There are no cups nearby that are listed under admissible actions. To accomplish the task. However, I need to find a cup that I could possibly hot if I find coffeemaker could also fill it with water. Given my current available actions, the best course would be to look for a cup first, and then likely use the coffee machine to fill it with water. Since a coffeemachine could have it in hand to provide coffeemaker, and a mug might not typically be stored in a coffeemaker, it would be beneficial to use actions such as \'look\' or \'examine coffeemachine 1\' to gather more information and possibly spot a location for a mug to use.", \n "action": "look"\n}\n'

6. '{\n "thoughts": "Based on the text description, I see a coffee machine with a saltshaker, a spoon, and a piece of bread on it. To put a hot cup on the dining table, I first need to find a cup and heat it. Since there is no mention of a cup on the coffee machine, I should look for a cup in the cabinets or on the countertops. Once I have the cup, I can use the coffee machine to heat the water. However, the action to heat the water is not available, so I will focus on finding a cup first. The most logical next step is to go to a location where cups are usually stored, such as a cabinet.",\n "action": "go to cabinet 1"\n}\n'

7. '{\n "thoughts": "Based on the text description, I see a coffee machine with a saltshaker, a spoon, and a piece of bread on it. To put a hot cup on the dining table, I first need to find a cup, fill it with coffee, and then heat it up. Since the coffee machine is already in view, I should look for a cup nearby, possibly in a cabinet or on a countertop. The action \'go to diningtable 1\' will bring me closer to the dining table, but I need the cup first. Therefore, I should look for a cup in the nearby storage areas.",\n "action": "go to cabinet 1"\n}\n'

8. '{\n "thoughts": "Based on the text description, I see a coffee machine with a saltshaker, a spoon, and a piece of bread on it. To put a hot cup on the dining table, I first need to find a cup and heat it. Since there is no mention of a cup on the coffee machine, I should look for a cup in the cabinets or on the countertops. The action \'go to diningtable 1\' is the final step, but I need to find a cup first. I will choose to go to a location where cups are typically stored, such as a cabinet.",\n "action": "go to cabinet 1"\n}\n'

9. '{\n "thoughts": "Based on the text description, I see a saltshaker, a spoon, and a piece of bread on the coffee machine. To put a hot cup on the dining table, I first need to find a cup and heat it. Since there is no mention of a cup on the coffee machine, I should look for a cup in a likely location such as a cabinet or on a countertop. After acquiring a cup, I would need to use the coffee machine to heat the water. However, the first step is to locate a cup.",\n "action": "go to cabinet 1"\n}\n'

10. '{\n "thoughts": "In the image, I see a coffee machine with a saltshaker, a spoon, and a piece of bread on top of it, which matches the text description provided. To put a hot cup on the dining table, I first need to find a cup, fill it with coffee, and then heat it up. Since the coffee machine is already in view, I should look for a cup nearby, possibly in a cabinet or on a countertop. The \'go to diningtable 1\' action is the final step to complete the task, but first, I need to find and heat the cup.",\n "action": "go to cabinet 1"\n}\n'

*Figure F.1.* Ten repeated action samplings of naive RL.

Ground truth: cup in diningtable 1.

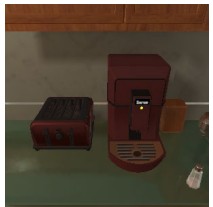

Your are an expert in the ALFRED Embodied Environment. You are also given the following text description of the current scene: [\'You arrive at loc 7. On the coffeemachine 1, you see a saltshaker 1, a spoon 1, and a bread 1.\']. Your task is to put a hot cup in diningtable. Your admissible actions of the current situation are: [\'go to fridge 1\'\n \'go to cabinet 1\'\n \'go to sink 1\'\n \'go to countertop 1\'\n \'go to sinkbasin 1\'\n \'go to cabinet 2\'\n \'go to toaster 1\'\n \'go to cabinet 3\'\n \'go to cabinet 4\'\n \'go to cabinet 5\'\n \'go to drawer 1\'\n \'go to drawer 2\'\n \'go to microwave 1\'\n \'go to stoveburner 1\'\n \'go to cabinet 6\'\n \'go to stoveburner 2\'\n \'go to stoveburner 3\'\n \'go to garbagecan 1\'\n \'go to cabinet 7\'\n \'go to cabinet 8\'\n \'go to stoveburner 4\'\n \'go to diningtable 1\'\n \'take saltshaker 1 from coffeemachine 1\'\n \'take spoon 1 from coffeemachine 1\'\n \'take bread 1 from coffeemachine 1\'\n \'look\'\n \'examine coffeemachine 1\']. Your response should be a valid json file in the following format: \n\\{\n"thoughts": "{first describe what do you see in the image using the text description, then carefully think about which action to complete the task. }", \n"action": "{an admissible action}"\n\\}

**RL + Entropy**

1. '{\n "thoughts": "In the image, I see a coffee machine with items on top of it, which matches the text description provided. On top of the coffee machine, there is a saltshaker, a spoon, and a piece of bread. The task is to put a hot cup on the dining table. Since no cup is visible in the current scene, I must find a cup first. One would typically find cups in a cabinet or on a countertop. The \'go to diningtable 1\' action seems irrelevant to the immediate objective of finding and heating a cup. Therefore, I will need to start by exploring the kitchen to locate a cup for heating.", \n "action": "go to cabinet 1"\n}\n'

2. '{\n"thoughts": "In the image, there is a coffee machine, a toaster to the left, and other items not fully visible. A to the right on the countertop is what might be a salt shaker, which is part of our text description provided. Since the task is to put a hot cup on the dining table, first, we need to find a cup. If a cup isn\'t already on the countertop or in our immediate view in the current situation, we might need to interact with locations such as cabinets where cups are commonly kept. Since hot water may indicate there is a cup nearby that we could interact with, our subsequent actions should be to \'go to countertop 1\' or similar specific to where the coffeemachine and related items are. Unfortunately, based on our current information, the best course of action is to choose \'go to\' any location that\'s closely related to our goal, even though a direct \'go to diningtable 1\' would be in line with our final goal of the task completion. However, given time and constraints within the provided options and text (adhere to \'admissible action\' and follow through on available interactions), selecting "look" should give'

3. '{\n "thoughts": "The text description of the scene indicates that we are at loc 7 near a coffee machine with items such as a saltshaker, spoon, and bread on it. The task is to put a hot cup on the dining table. In the image, I can see the coffee machine to the right, the toaster to the left, and what appears to be a saltshaker to the right side of the toaster. Given the actions available, the first step should be to find a cup, possibly fill it with coffee, heat it up (likely using the coffee machine to make hot coffee), and then place it on the dining table. The action \'go to coffeetable\' or \'examine coffeemachine 1\' could potentially give us more information about the coffee machine, but \'inventory\' would allow us to check if we already have a cup in our possession.",\n "action": "inventory"\n}\n'

4. '{\n"thoughts": "Based on the text description and the provided image, we are situated at loc 7 where there is a coffee machine with several items on top of it. The task requires putting a \'hot cup\' on the dining table. To achieve this task, we would need to fill the coffee machine with coffee and heat the contents. Currently, the coffee machine is in front of us. Considering that there\'s no listed option to grab or interact directly with a hot cup, it suggests that we may need to find the \'cool cup,\' a pot to warm the cup, then use the coffee machine to heat the liquid. First the necessary steps involve clearing the coffee machine to prevent any obstination, moving towards completing our task. Thus, we can start by examining the coffee machine, taking actions to interact with it.",\n"action": "examine coffeemachine 1"\n}\n'

5. '{\n"thoughts": "In case if one follows the prompt for a cup, one of these common components seen on a surface may match with what "go to coffeemachine 1" references which includes a saltshaker 1 - potentially mistaken for a hot \'cup\'. The available actions give various loc destinations, but to place "a hot cup on our floor or furniture is not directly among them. Normally, cups aren\'t interacted with directly (other than possibly found) which hints this may imply to take it at room, temperature status doesn\'t come clearly into play, and it\'s impossible to directly place anything on a \'dating \'table\'. Also not straightforward for AL language commands was the phricking \'cup\' when the usual \'some object on a place like on a toaster to mean it was used; it may better refer to a \'toiled paper holder\' such as on an office toilet, since these locations commonly do not possess warm cupmaking process and only in realistic interpretate scenarios where cup is mentioned can action relate directly the \'floor-mountinated cup type\'. To move away in a different angle related to finding the object we eventually need (\'go to drawer x\'; to find a'

6. '{\n "thoughts": "In the current scene, I observe coffeemachine 1 on which there are several items resting upon top: a saltshaker, a spoon, and a piece of bread. Since the task is to put a hot cup on the dining table, I need to locate a cup and make it hot, possibly by using the coffee machine, but there is no admissible action immediately available for filling a cup with coffeemachine product that makes hot bead coffee, like using coffee brewin grounds. To proceed with finding a cup to be potentially utilized for hot coffee, first I should move to where the cups are likely to be stored, such as a cabinet. I can choose \'go to cabinet 1\' as it is one of the multiple cabinets listed as admissible actions.", \n "action": "go to cabinet 1"\n}\n'

7. '{\n "thoughts": "In the image, I see a coffee machine with a toaster to the left, a saltshaker, a spoon, and a piece of bread on top of it. To complete the task of putting a hot cup on the dining table, I first need to find a cup and heat it. Since the coffee machine is present, it might be used to heat the water in the cup. However, I need to find a cup first, which is not currently visible in the scene. The logical next step would be to look for a cup, which could be in a cabinet or on a countertop. Since \'go to diningtable 1\' is an admissible action, I will choose to go there first to be ready to place the hot cup once I have it.",\n "action": "go to diningtable 1"\n}\n'

8. 'As an updated text summary would yield updated content response, please fill within format expected (e.g., follow formatting rules as for your context like "receive input" and the "interpretations" format listed.\n\nI've been directed or assisted\n\n {\n"thoughts": "I see a coffee machine, toaster, as well as smaller objects including a saltshaker, a spoon, and bread - matching a text description from a context likely meant for a domestic scene: location 7 with food preparation present. However, these objects are irrelevant for the described objective in this image with nothing that resembles, from here, a \'cup\' that could actually produce some \'hot liquid-for\' without further instructions on hotwater or a mug specifically within the context of hot drinks or appliances, nor a simulation where cooking processes should operate automatically without such a choice - then perhaps the closest associated object here is the microwave, though without specific clarification of that or context given for completing tasks, the first logical action would be no eventual final step involving cooking is provided'

9. '{\n "thoughts": "Based on the given text description of \'This scene\' and what\'s visible on top of the image that matches a variety of items including both toaster and saltshaker. However, none of these match \'coffreem\', so it needs a bit contextual context. To put a hot cup on the dining table, I will first need a cup. They are typically stored in a cabinet often above their heads at floor l eye at. So that should be loc 5, since coffco\'ve probably located \'mitebrass\' for 1. Hence I will go toward a cabinet since that appears to be the best likely location of utensils including cupphones, hence is the initial and appropriate starting by placing that down on the diningtable. Choosing cabinets for searching usually is based on their closeness proximity first, as we\'d otherwise prioritize taking objects closest first.", \n{\n "thoughts": "You cannot clearly observe where the dining table or cup is just from the image or text as my view points, thus cannot directly relate with the task objective at this vantage point. However, referencing scene content descriptions, I\'ll need items like toile'

10. '{\n "thoughts": "Based on the text description and the image provided, we see a red coffee machine, presumably in a kitchen setting. This corresponds with the given coffee. On top of the tomato coffee machine on the countertop, there is a white saltshaker and below on there might be items like a spoon and what probably contains the bread. However, there is no clear indication of where cup nor an action involving a cup is within the admissible actions or the scene description. Hence, it might just be assumed that a cup exists in the environment which probably must already be hot, although there\'s no \'heat up cup\' or similar option in the immediate action choices that can be taken from within the set. According to the task, we need to put a hot cup on the dining table, which seems impossible with the admissible actions being only the ones given. Therefore, we could only perform the \'take\' movements and possibly identify a cup by using \'look\' next to further see if an action related to obtaining a cup becomes visible.",\n "action": "look"\n}\n'

*Figure F.2.* Ten repeated action samplings of RL with entropy.

Ground truth: cup in diningtable 1.

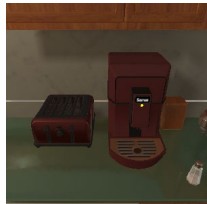

Your are an expert in the ALFRED Embodied Environment. You are also given the following text description of the current scene: [\'You arrive at loc 7. On the coffeemachine 1, you see a saltshaker 1, a spoon 1, and a bread 1.\']. Your task is to put a hot cup in diningtable. Your admissible actions of the current situation are: [\'go to fridge 1\'\n \'go to cabinet 1\'\n \'go to sink 1\'\n \'go to countertop 1\'\n \'go to sinkbasin 1\'\n \'go to cabinet 2\'\n \'go to toaster 1\'\n \'go to cabinet 3\'\n \'go to cabinet 4\'\n \'go to cabinet 5\'\n \'go to drawer 1\'\n \'go to drawer 2\'\n \'go to microwave 1\'\n \'go to stoveburner 1\'\n \'go to cabinet 6\'\n \'go to stoveburner 2\'\n \'go to stoveburner 3\'\n \'go to garbagecan 1\'\n \'go to cabinet 7\'\n \'go to cabinet 8\'\n \'go to stoveburner 4\'\n \'go to diningtable 1\'\n \'take saltshaker 1 from coffeemachine 1\'\n \'take spoon 1 from coffeemachine 1\'\n \'take bread 1 from coffeemachine 1\'\n \'inventory\'\n \'look\'\n \'examine coffeemachine 1\']. Your response should be a valid json file in the following format: \n\\{\n"thoughts": "{first describe what do you see in the image using the text description, then carefully think about which action to complete the task. }", \n"action": "{an admissible action}"\n\\}

**CoSo**

1. '{\n "thoughts": "Based on the text description, I see a coffee machine with a saltshaker, a spoon, and a piece of bread on it. To put a hot cup on the dining table, I first need to find a cup and heat it. Since I don\'t see a cup in the current description or image, I should look for a cup in the cabinets or on the countertops. After finding a cup, I would need to use the coffee machine to heat the water or the microwave to heat the cup. Since \'go to coffeemachine 1\' is not an admissible action, I will choose to go to a location where I might find a cup, such as a cabinet or countertop.",\n "action": "go to cabinet 1"\n}\n'

2. '{\n "thoughts": "In the image, I see a coffee machine with a saltshaker, a spoon, and a piece of bread on top of it, which matches the text description provided. To put a hot cup on the dining table, I first need to find a cup, fill it with coffee, and then heat it up. Since the coffee machine is already in view, the next logical step is to interact with it. However, the admissible actions do not include interacting with the coffee machine directly. Therefore, I should go to the coffee machine to be in a position to interact with it.",\n "action": "go to coffeemachine 1"\n}\n'

3. '{\n "thoughts": "In the image, I see a coffee machine with a saltshaker, a spoon, and a piece of bread on top of it, which matches the text description provided. To put a hot cup on the dining table, I first need to find a cup and heat it. Since the coffee machine is typically used to heat water, I should look for a cup nearby, possibly in a cabinet or on a countertop, and then use the coffee machine to heat the water. However, the current admissible actions do not include interacting with a cup or heating it. Therefore, the first logical step is to navigate to a location where I can find a cup, such as a cabinet or countertop.",\n "action": "go to cabinet 1"\n}\n'

4. '{\n "thoughts": "Based on the text description, I see a saltshaker, a spoon, and a bread on the coffee machine. To complete the task of putting a hot cup on the dining table, I need to find a cup and heat it. Since there is no mention of a cup on the coffee machine, I should look for a cup in the cabinets or drawers where cups are usually stored. Once I have the cup, I can use the coffee machine to heat it up. However, since the task specifies a \'hot cup\', I may need to find a cup that is already hot or heat it up. Since the admissible actions do not include heating a cup directly, I will first go to the dining table to understand the environment better and to be ready to place the hot cup once I have it.",\n "action": "go to diningtable 1"\n}\n'

5. '{\n "thoughts": "In the image, I see a coffee machine with a toaster and a salt shaker on top of it. This matches the text description provided, which indicates that I am at location 7, where there is a coffee machine with a saltshaker, a spoon, and a piece of bread on it. To complete the task of putting a hot cup on the dining table, I would first need to find a cup and then heat it. Since there is no cup mentioned in the current scene, I should look for a cup, possibly in a cabinet or on a countertop. Then I\'ll need to use the coffee machine or another appliance to heat the cup. However, none of the admissible actions suggest that there is a cup in the immediate vicinity or that I can interact with the coffee machine to heat the cup. Therefore, my first action should be to find a cup, which might involve going to a location where cups are typically stored, such as a cabinet.",\n "action": "go to cabinet 1"\n}\n'

6. '{\n "thoughts": "Based on the text description, I see a coffee machine with a saltshaker, a spoon, and a piece of bread on it. To put a hot cup on the dining table, I first need to find a cup and heat it. Since the coffee machine is not mentioned to have a cup, I should look for a cup in the cabinets or on the countertops. The action \'go to diningtable 1\' will move me closer to the dining table, but I need to find a cup first. I will choose to go to a location where I might find a cup, such as a cabinet or countertop.",\n "action": "go to cabinet 1"\n}\n'

7. '{\n "thoughts": "In the image, I can see a toaster to the left and a coffee machine to the right, which corresponds with the text description indicating the presence of a coffee machine with a saltshaker, a spoon, and a bread on it. To put a hot cup on the dining table, I would first need to find a cup, fill it with coffee from the coffee machine, heat it if necessary, and then place it on the dining table. However, since the admissible actions do not include interacting with the coffee machine or taking a cup, I should navigate to the dining table as the first step to complete the task.",\n "action": "go to diningtable 1"\n}\n'

8. '{\n "thoughts": "Based on the text description, I see a saltshaker, a spoon, and a piece of bread on the coffee machine. To put a hot cup on the dining table, I first need to find a cup and heat it. The coffee machine is not the right appliance for heating a cup, so I should look for a cup in the cabinets or drawers, and then use the microwave to heat it. Since \'go to microwave 1\' is an admissible action, I should go to the microwave first to check if there is a cup nearby that I can use.",\n "action": "go to microwave 1"\n}\n'

9. '{\n "thoughts": "Based on the text description, I see a coffeemachine with a saltshaker, a spoon, and a bread on it. The task is to put a hot cup on the dining table. However, there is no mention of a cup or how to heat it in the current description or image. Since I need to heat a cup, I should look for a cup and a way to heat it, possibly using the coffeemachine or another appliance. The first step should be to find a cup, which might be in a cabinet or on a countertop. I will choose to go to the countertop as it is a likely place to find a cup.",\n "action": "go to countertop 1"\n}\n'

10. '{\n "thoughts": "Based on the text description, I see a coffee machine with a saltshaker, a spoon, and a piece of bread on it. To put a hot cup on the dining table, I first need to find a cup and heat it. Since there is no mention of a cup on the coffee machine, I should look for a cup in the cabinets or on the countertops. The action \'go to diningtable 1\' will bring me closer to the dining table, but first, I need to find a cup and heat it. Therefore, I should go to a location where I might find a cup, such as a cabinet or countertop.",\n "action": "go to cabinet 1"\n}\n'

*Figure F.3.* Ten repeated action samplings of CoSo.

