# OpenReview forum: "Towards Efficient Online Tuning of VLM Agents via Counterfactual Soft Reinforcement Learning"
_ICML.cc/2025/Conference — ICML 2025 poster_

### Official Review · Reviewer_vQ7r · 2025-03-07

**Overall Recommendation:** 4

**Summary:**

This paper keenly and ingeniously identifies that the influence of tokens on the parsed action varies, with a small subset of action-critical tokens decisively shaping the final outcome. Therefore, after calculating causal weights using Structural Causal Models (SCM) and counterfactors, the authors propose Counterfactual Soft Reinforcement Learning (RL). The entire paper is easy to follow and contains core insights.

**Claims And Evidence:**

The experiments in this paper effectively validate the effectiveness of Coso without falling into the trap of overclaiming.

**Essential References Not Discussed:**

No

**Experimental Designs Or Analyses:**

The paper conducted experiments on RL4VLM, ALFWorld, and AitWorld, demonstrating the effectiveness of the proposed method. Furthermore, it largely adheres to the original settings of these benchmarks, making the results highly credible.

**Methods And Evaluation Criteria:**

The methodology in this paper is highly reasonable under the guidance of its core insight. It first employs Structural Causal Models (SCM) to predict causal weights, which are then used to enhance the Soft-RL objective, effectively coupling it with these causal weights. However, there are a few minor issues regarding the prediction of causal weights that need to be addressed.

+ **What specifically is the distance metric?** Could you provide some examples for the following three experiments? For instance, in the Game Card experiment, what does the sentence look like after removing a certain token, and what function is used to measure the distance metric compared to the original?

+ **Regarding the replacement of tokens with a nullified value** why not randomly select a similar token instead? This actually depends on how the parse_action function is implemented. For example, in the ALF or Android experiments, the parse_action function may require adherence to a specific pattern. If an action is represented as (A, B), A and B are typically more critical. However, if tokens are randomly replaced, it could lead to violations of this pattern, causing parse_action to fail and return a large negative reward.

+ **Speed Problem** Does the calculation of causal weights take a long time, and could it become a speed bottleneck in the entire optimization pipeline?

**Other Comments Or Suggestions:**

Please refer to method part.

**Other Strengths And Weaknesses:**

Please refer to method part.

**Questions For Authors:**

Please refer to method part.

**Relation To Broader Scientific Literature:**

No

**Theoretical Claims:**

The paper provides a detailed explanation of the theory, and the supplementary material includes thorough proofs of the theory. Therefore, there are no issues with the theoretical part.

---

> ### Author Rebuttal · Authors · 2025-03-31
>
> Thank you very much for your thoughtful review. We sincerely appreciate you pointing out our method's core insight, theoretical soundness, and the credibility of our experimental evaluation. Below, we provide our detailed responses to your remaining questions.
>
> > Q1. What specifically is the distance metric? Examples? What function is used to measure the distance?
>
> - (Distance metric) As defined in Eq. (6), the distance metric is the **absolute difference in action likelihood between the original input and a modified input**, where a specific token is replaced by a placeholder (e.g., "pad_token").
> - (Example) Take the Game Card as an example.
>   - The original input $y$:
>   `{"cards": [2, 6], "formula": "2*", "thoughts": "'2*' is an incomplete formula. Since '2*6=12', I should append '6' to the current formula", "action": "6"}`
>   We first feed it into the SCM and let SCM output the baseline action likelihood $\mathbb{P}(a|y)$. Then, we remove each token by replacing it with a placeholder. For example:
>   1. Remove 1st token $y^{-1}\cup y^1_{\text{null}}$:
>   `<pad>"cards": [2, 6], "formula": "2*", "thoughts": "'2*' is an incomplete formula. Since '2*6=12', I should append '6' to the current formula", "action": "6"}`
>   2. Remove 2nd token $y^{-2}\cup y^2_{\text{null}}$:
>   `{<pad>cards": [2, 6], "formula": "2*", "thoughts": "'2*' is an incomplete formula. Since '2*6=12', I should append '6' to the current formula", "action": "6"}`
>   3. Remove 3rd token $y^{-3}\cup y^3_{\text{null}}$:
>   `{"<pad>": [2, 6], "formula": "2*", "thoughts": "'2*' is an incomplete formula. Since '2*6=12', I should append '6' to the current formula", "action": "6"}`
>   4. ...
>   Each modified sequence is then fed to SCM to obtain a new action likelihood $\mathbb{P}(a|y^{-i}\cup y^i_{\text{null}})$ for $i=1,2,...,n$.
> - (Function) **The distance function is given by $D=|\mathbb{P}(a|y)-\mathbb{P}(a|y^{-i}\cup y^i_{\text{null}})|$**. This measures the absolute change in action likelihood caused by nullifying a specific token $y^i$. By observing how each token's removal changes (or does not change) the final action likelihood, we can measure each token's causal importance.
>
>
> > Q2. Why not randomly select a similar token instead? Causing parse_action to fail and return a large negative reward?
>
> Thank you for raising this point. If we understand correctly, your concern is that using a nullified value might cause the parse_action function to fail, leading to invalid actions and large negative rewards during the interaction between the agent and the environment.
>
> We would like to clarify that our method **does not have this issue** because the entire causal computation process takes place after the agent-environment interaction phase. Specifically:
> - As shown in Algorithm 1 (Lines 6–11), the rollout phase remains exactly the same as in a standard RL loop. In this step, **the agent interacts normally with the environment** and collects trajectory data (without involving the SCM or any token nullification).
> - After that (Lines 12-18 of Algorithm 1), token nullification and the computation of causal weights $B_{y_t\rightarrow a_t}$ are conducted **offline**, using the stored transitions $(s_t, a_t, r_t, y_t)$ in the replay buffer. Since both the action $a_t$ and reward $r_t$ **have already been determined and logged** at that point, the SCM's token nullification analysis does not affect action parsing or the reward signal in any way.
>
> > Q3. GPU memory usage and training time
>
> We have reported the computational budget of causal weights in **Appendix C**. It adds only 0.01 B parameters (<0.2\%), 0.7 GB of GPU memory (<2\%), and 0.5 H100 GPU hours (<4\%), which are modest and introduce small overhead.
>
> Thank you again for your valuable feedback. We will incorporate clarifications on the above three points in our revised version.

---

### Official Review · Reviewer_G6Px · 2025-03-10

**Overall Recommendation:** 4

**Summary:**

This paper introduces CoSo, a soft reinforcement learning (RL) method for fine-tuning Visual Language Models (VLMs). CoSo incorporates a per-token weighted entropy regularization term, encouraging exploration on impactful tokens. It is built on two key contributions:
- A counterfactual approach, where generated tokens are replaced to assess their impact on the final environment action.
- An adaptation of the soft RL objective, in which each token’s entropy is weighted based on its importance.

Empirical results on standard VLM-based agent benchmarks demonstrate that adding CoSo to AWR or PPO significantly improves performance. Ablation studies further highlight that per-token entropy weighting accelerates exploration by focusing on key tokens.

## update after rebuttal
Most of my concerns below were related to a lack of explanations, especially regarding the method. The authors' response clarified all of my concerns, as well as my misunderstandings (e.g., concerning additional environment interactions induced by the method).

Additionally, my initial review proposed adding new experiments on LLMs (and not only on VLMs), which I believe could have a high impact. During the rebuttal period, the authors conducted such experiments, highlighting the efficiency and potential of their method on LLMs as well. I am therefore recommending acceptance of this paper.

**Claims And Evidence:**

The authors highlight and tackle a key problem in applying RL to LLMs and VLMs: exploration. While properly exploring remains a key challenge in RL in general, classic methods mostly relying on random action selection fall short when actions are natural language sentences. Indeed, the action space is huge, most token sequences are not meaningful sentences, and all tokens do not have the same importance. Such work is, therefore, timely.

The introduced method makes little changes to the classic soft RL objective. It is, therefore, easily applicable to existing methods while showing significant improvement.

**Essential References Not Discussed:**

There are no essential references not discussed.

**Experimental Designs Or Analyses:**

The benchmark used, and the baselines provided seem like natural choices. The analyses are insightful. I particularly enjoyed Section 5.3, which highlights CoSo's efficiency. Section 5.2 could benefit from quantitative results studying the tokens with high weight (e.g., the ones in Figure D.2). The authors also fairly showed the additional computational cost introduced by CoSo in Appendix C.

**Methods And Evaluation Criteria:**

As mentioned in my summary, CoSo relies on two main components. The weighted entropy part is well-discussed, and the experiments (especially Section 5.3) show its efficiency. However, the implementation of the "Counterfactual Reasoning" clearly lacks details (which I did not find in the appendices), especially considering how central this part is in CoSo:
- The authors used an additional (smaller-scale) model to predict the likelihood of an environment action based on a token sequence. Almost no motivation is provided for this choice.
- Appendix B.1 indicates that this model is trained offline and online. How are the data collected for the offline training?
- The paper says a CrossEntropy loss is used to train the model online. What is the ground truth distribution used in the loss?
- What is given as input to the model?
- How accurate was it? Could an even smaller model be used?
- What does it mean to "nullify" a token?
- The conclusion mentions up to 300 tokens per action. Is it necessary to "nullify" each token?
Did I understand correctly that, at each step, the method requires as many environment interactions as the number of tokens (plus one to actually play the action) in the chosen action to evaluate each token's importance? If so, this should be mentioned.

I would also like to mention that CoSo does not really seem specific to VLMs: the exploration challenge when using natural language actions also applies to LLM-based agents, for instance. I think this paper would have an even greater impact if the authors added experiments on benchmarks for LLM agents such as the ones in POAD (Wen et al., 2024) or LOOP (Chen et al., 2025).

**Other Comments Or Suggestions:**

There is a typo int he title of Appendix C: the "t" at the end of "budget" is missing.

The paper also repeatedly states that prior work "rely on classic trial-and-error exploration" (for instance, at the end of the second paragraph in Section 2). All RL approaches, including CoSo, rely on trial-and-error exploration. What differs in CoSo is that it prioritizes exploration over tokens. I think this could be made clearer.

In the second paragraph of Section 3, the VLM-based policy outputs a distribution over environment actions ($\pi_{\theta}(a|s)$), while in the third paragraph, it outputs a distribution over tokens. As the function mapping a sequence of tokens to an environment action has already been defined in the first paragraph, I think it would be clearer to consider the policy over tokens in the second and third paragraphs.

Finally, the beginning of Section 5 explains that the experiments provide an analysis of CoSo's computational budget, which is currently in appendices and never referenced in the main paper.

**Other Strengths And Weaknesses:**

The lack of details on the implementation of the counterfactual reasoning is the main reason for me to recommend only a "Weak accept". I am willing to increase my score if the authors provide details that would be added to the manuscript.

As I also mentioned in my review, extending the paper to LLM agents would have a very impactful effect.

**Questions For Authors:**

I do not have any further questions.

**Relation To Broader Scientific Literature:**

The "Counterfactual reasoning" component of CoSo proposes a method for measuring the impact of a token in selecting an action in the environment. While not exactly the same, it seems closely related to token-level credit assignment, which has been studied in prior work, particularly in POAD (Wen et al., 2024). This link should be discussed.

**Theoretical Claims:**

The authors provide a theoretical analysis of CoSo, showing its soundness as a soft RL objective.
I only checked the proof in A.1, but the claims appear pretty straightforward.

---

> ### Author Rebuttal · Authors · 2025-03-31
>
> Thank you very much for your detailed and thoughtful review. We truly appreciate you highlighting the strengths of our method and thinking it timely, well-discussed, easily applicable, and supported by insightful experiments. Regarding your questions, we provide our responses as follows.
>
> > Q1. Why a smaller model?
>
> We chose a smaller model mainly to **reduce computational cost and training time**. As shown in Appendix C and D, we found that the lightweight model already works well for SCM while introducing small overhead.
>
> > Q2. How is the offline data collected?
>
> The offline data used to train the SCM **comes from the SFT dataset in the original paper** (DigiRL and RL4VLM), which includes the agent's "observations" and its "text responses". For SCM's training, we only use the "text responses" as inputs and their corresponding action labels as ground truth.
>
> > Q3. The ground truth in the CrossEntropy loss.
>
> In our implementation, the SCM is trained as an action classifier. Thus, **the ground truth is the label of the parsed action** corresponding to the agent's text output.
> For example, SCM input: "`Action Plan: [PRESS_HOME,DUAL_POINT, DUAL_POINT,DUAL_POINT] ; Action Decision: "action_type": ...`", SCM ground truth: `2` (assuming `PRESS_HOME` maps to index 2)
>
> > Q4. What is the input of SCM?
>
> - As explained above, the input is the **agent's text output $y$** (e.g., `Action Plan: [PRESS_HOME,DUAL_POINT, DUAL_POINT,DUAL_POINT] ; Action Decision: "action_type": ...`), then SCM predicts its correct environment action category.
> - Moreover, when computing causal weights, we also nullify a specific token (e.g., the first token ($y^{-1}\cup y^1_{\text{null}}$)) and **feed the modified sentence** "`<pad> Plan: [PRESS_HOME,DUAL_POINT, DUAL_POINT,DUAL_POINT] ; Action Decision: "action_type": ...`" into the SCM to get token's causal influence.
>
> > Q5. How accurate was SCM? Could a smaller model be used?
>
> - During the training, the SCM achieves ~100% accuracy in AitW, ~90–100% in Gym Cards, and ~70–80% in ALFWorld.
> - Yes, the smaller model can be used, especially in high-accuracy tasks. However, there is a trade-off between size and the quality of causal weights.
>
> > Q6. What does "nullify" a token mean?
>
> To "nullify" a token means **replacing it with a placeholder token (e.g., `pad_token` or `unk_token`)** to simulate its absence.
>
> > Q7. Is it necessary to "nullify" each token? Does it require extra environment interactions?
>
> Yes, we **need to nullify each token** to compute its causal importance. However, this process requires **no extra environment interactions**.
> - As shown in Alg. 1 (Lines 6–11), the 'Rollout phase' remains exactly the same as in a standard RL loop. In this step, the agent interacts normally with the environment and collects trajectory data **(without the SCM or token nullification)**.
> - After that, we evaluate the token's importance using the stored transitions $(s_t, a_t, r_t, y_t)$ in the replay buffer $\mathcal{U}$. Specifically:
>     1. Feed the raw $y_t$ into the SCM and let SCM output the baseline action likelihood $\mathbb{P}(a_t|y_t)$.
>     2. Nullify a specific token and feed $y_t^{-i}\cup y^i_{\text{null}}$ obtain the action likelihood $\mathbb{P}(a_t|y_t^{-i}\cup y^i_{\text{null}})$.
>     3. Compute token's importance via $\mathcal{B}^i_{y_t\rightarrow a_t}=|\mathbb{P}(a_t|y_t)-\mathbb{P}(a_t|y_t^{-i}\cup y^i_{\text{null}})|$.
> - The above process **relies only on the replay buffer $\mathcal{U}$ and the SCM model (without environment interactions)**. Thus, environment interaction cost stays the same.
>
> > Q8. Extension to LLM agents
>
> Thank you for your valuable suggestion. We totally agree that CoSo is not limited to VLM agents and it applies naturally to LLM agents operating in purely textual environments.
> To show this, we include an experiment on `AlfredTWEnv`, a purely text-based benchmark in the ALFWorld where several LLM agents (e.g., ReAct, Reflexion) have been evaluated. We achieve RL4VLM and CoSo on `Qwen2.5-7B-Instruct` and train the LLM agents from scratch (without SFT) over 12,000 environment steps. Here's the result:
>
> |AlfredTWEnv|Pick|Look|Clean|Heat|Cool|Pick2|Avg.|
> |-|-|-|-|-|-|-|-|
> |RL4VLM (LLM Agent)|62.9|**38.5**|35.0|33.3|25.9|**11.1**|32.8|
> |CoSo (LLM Agent)|**77.1**|24.2|**40.7**|**37.5**|**35.3**|7.0|**39.6**|
>
> > Reference
>
> Thank you for the suggestion. We'll cite and discuss token-level credit assignment works like POAD in the revised Related Work section.
>
> > Writing and presentation
>
> Thank you for your helpful editorial suggestions. We'll fix the typos in the title of Appendix C, clarify the use of "trial-and-error", unify the policy definition over tokens in the second and third paragraphs in Section 3, and refine the beginning of Section 5 by adding a pointer to Appendix C.
>
> Thank you again for all your constructive feedback. We will include these clarifications, especially the counterfactual reasoning, and updates in our revision.

---

> > ### Comment · Reviewer_G6Px · 2025-04-02
> >
> > I thank the authors for their reponse.
> > They clarified all my concerns as well as my missunderstandings (e.g. concerning addititional environment interactions induced by the method). I also deeply appreciated the additional experiments using LLMs, which I believe can have a high impact.
> >
> > Consequently, I will raise my score.
> >
> > I also have a question regarding these additional experiments: If I recall well, RL4VLM first applies SFT to the VLM before using RL. You said you trained the LLMs from scratch, is it both for RL4VLM and CoSo?

---

> > > ### Author Response · Authors · 2025-04-02
> > >
> > > Thank you very much for raising your score! We are glad to answer your remaining question and would appreciate any further response.
> > > > Is it both for RL4VLM and CoSo?
> > >
> > > Yes — in the pure-text `AlfredTWEnv` shown in the rebuttal, we trained **both** RL4VLM and CoSo from scratch. This was mainly due to the following reasons:
> > > 1. The SFT dataset used in RL4VLM for ALFWorld contains **both image and text modalities**, making it unsuitable for directly fine-tuning LLM agents. Simply dropping the image input would result in incomplete environment information for the agent.
> > > 2. Collecting a new pure-text SFT dataset specifically for the LLM agent would have been quite **time-consuming**, especially within the short rebuttal period.
> > > 3. The pure-text environment is **much simpler** compared to the multimodal one, and the LLM agent (`Qwen2.5-7B-Instruct`) **already has a good initialization**, so training from scratch worked well in this case.
> > >
> > > We hope this clarifies the experimental setup.

---

### Official Review · Reviewer_igVU · 2025-03-14

**Overall Recommendation:** 3

**Summary:**

This paper investigates the fine-tuning of VLM agents through a two-stage offline-to-online process, with a particular focus on the online phase, termed CoSo. CoSo uses soft Q-learning to improve exploration within sequential reasoning frameworks, such as chain-of-thought (CoT). The entropy term is computed by summing the entropy values across output distributions. Unlike prior methods, CoSo incorporates causal-aware entropy values, derived using causal weights from a structured causal model (SCM). Specifically, the SCM assesses action distribution change for counterfactual token sequences, which are generated by nullifying a single token. A significant change in action likelihood suggests that the nullified token is essential for decision-making. As a result, CoSo assigns weights to each entropy value proportional to the likelihood change caused by nullifying the corresponding token. Experimental results show that CoSo consistently outperforms baselines across various VLM agent benchmarks, confirming the algorithm's effectiveness.

**Claims And Evidence:**

The use of soft RL to enhance exploration is well-motivated and convincing, supported by its proven theoretical and empirical effectiveness in traditional RL. In contrast, although the causal weighting of entropy terms is intuitive and demonstrates empirical effectiveness, it lacks a clear theoretical foundation or logical justification to establish a strong correlation between the causal weights and the significance of the tokens. While the concept of importance weighting is compelling in principle, the proposed methodology, which employs causal weights with the given formulation, would benefit from further verification.

**Essential References Not Discussed:**

One of the key components of CoSo is the structured causal model (SCM), but there is no citation related to this. Also, such SCM is BERT-based, thus BERT paper should be referenced.

**Experimental Designs Or Analyses:**

I reviewed most aspects of the experimental design and analysis, including the experimental setups and ablations. Since CoSo is implemented on top of the baseline, using the same hyperparameters as the baseline for the corresponding components appears to be a fair approach. Additionally, conducting quantitative analyses, such as token generation visualization and causal weight heatmaps, significantly enhances the interpretability of the results. However, the ablation studies could be further refined. A hyperparameter sensitivity analysis, particularly with respect to the learning rate in the SCM or the entropy coefficient, could provide additional insights into the robustness of the algorithm. Furthermore, exploring alternative weighting strategies, such as attention weights or approaches similar to the Shapley value, could offer valuable comparisons. Please refer to the questions for more details.

**Methods And Evaluation Criteria:**

CoSo is applied on top of existing RL-based online VLM fine-tuning algorithms, implying high expandability and flexibility. The benchmarks, which consist of AitW, Gym Cards, and ALFWorld, are well-regarded in evaluating VLM agents and span various domains of VLM decision-making.

**Other Comments Or Suggestions:**

1. For eq. 6, it would be preferable to use a single vertical line to denote absolute value, rather than a double vertical line.

**Other Strengths And Weaknesses:**

N/A

**Questions For Authors:**

1. For causal weights, we can use a likelihood difference of all actions, for instance, divergences, instead of considering only a likelihood difference from the chosen action. Can you provide the results?

2. It seems that there is no ablation study for the learning rate in the SCM or the entropy coefficient. Can you provide the results, particularly the latter?

3. As observed in Appendix D, the model seems to treat intermediate reasoning components inconsistently, sometimes overlooking them and at other times considering them important. What are underlying tendencies that might explain this phenomenon?

4. Is there any rationale for using causal weights to weight entropy values, other than the ones provided in the paper?

5. Instead of calculating causal weights, another simpler approach might be to use attention weights. Can you provide the results using them?

6. The methodology used for calculating causal weights is based on the leave-one-out (LOO) scheme, which is often compared to the Shapley value. Can you provide the results using the Shapley value or its variants, such as SHAP?

**Relation To Broader Scientific Literature:**

Since VLM agents can be applied to various real-world scenarios such as web search and task automation, the ability of CoSo to enhance the performance of VLM agents without requiring a larger model or extensive training resources is noteworthy. This suggests that CoSo might enable the use of smaller VLM models while maintaining satisfactory performance, potentially improving accessibility and efficiency.

**Theoretical Claims:**

I reviewed all aspects, including policy evaluation, policy improvement, and policy iteration for the framework used in CoSo. While there are no critical flaws, I noticed a few minor issues, such as unexplained shorthand notations (ex) A.6, A.11) and some typos (ex) A.7).

---

> ### Author Rebuttal · Authors · 2025-03-31
>
> We sincerely appreciate your in-depth and constructive feedback. We are also grateful for kindly pointing out that our method is well-motivated and convincing, compelling in principle, and implies high expandability and flexibility. Regarding your concerns, we provide our responses below:
>
> > Q1: The results of using the likelihood difference of all actions.
>
> We tried both **KL divergence and L2 distance** on Gym Cards to take all actions into account. Both produced **similar results** in terms of causal weights and overall performance. KL divergence is asymmetric and more sensitive to small distributional differences, so it may require more careful smoothing or regularization.
>
> ||KL Divergence|L2 Distance|Ours|
> |-|:-:|:-:|:-:|
> |NL|98.5|100.0|100.0|
> |BJ|40.3|41.8|41.5|
>
> > Q2: Ablation study for the learning rate and entropy coefficient.
>
> We provide ablations about the learning rate and the entropy coefficient on the Gym Cards NL task below. We found that CoSo **needs an appropriate learning rate and entropy coeff to work best**. For example, if entropy coeff is too large, it excessively perturbs the output distribution, reducing the agent's stability; if too small, it approximates a setting without exploration incentives.
>
> |Learning Rate|Result|\||Entropy Coeff|Result|
> |:-:|:-:|:-:|:-:|:-:|
> |1e-3|96.3|\||10.0|78.8|
> |1e-4|100.0|\||1.0|100.0|
> |1e-5|100.0|\||0.1|98.3|
> |1e-6|94.3|\||0.01|95.5|
> |1e-7|90.8|\||0.001|88.5|
>
> > Q3: Causal differences across intermediate reasoning components.
>
> Actions like `DUAL_POINT` and `TYPE` typically involve **more complex token patterns** than simpler actions such as `PRESS_HOME` or `PRESS_ENTER`. Beyond the main action token, they often include **additional elements (e.g., coordinate values or type-specific content) that vary across instances**. These extra tokens can introduce nuanced information, making the SCM more **sensitive** to intermediate components.
>
> > Q4: Rationale for using causal weights to weigh entropy values?
>
> **Yes, the theory of weighted entropy has been formally studied in [1]**: $h_\varphi(p)=-\sum_i\varphi(x_i)p(x_i)\log p(x_i)$ (see Eq.(1.2) in [1]). Here $\varphi(x_i)$ is a non-negative weight function, presenting a value/utility of the outcome $x_i$. This formulation makes entropy context-dependent, assigning different weights based on the value of each outcome. In our case, we adopt causal weights as the utility function, assigning higher weights to more important tokens within an action.
>
> [1] Kelbert, Mark, Izabella Stuhl, and Yuri Suhov. "Weighted entropy: basic inequalities." Modern Stochastics: Theory and Applications 4.3 (2017): 233-252.
>
> > Q5 & Q6: Attention weights and Shapley value.
>
> Thank you for your valuable suggestions. We evaluated the attention weight and the Shapley value and presented the results below.
> - (Attention weights) We found that attention weights **perform suboptimally**. Since attention weights dynamically evolve during training, especially early stage of the training, they often lack stability and fail to consistently reflect token importance.
> - (Shapley value) The computation of Shapley value scales exponentially with token count ($2^n$) which is computationally infeasible. Thus we use Monte Carlo sampling with 10 subsets. It produces **similar results as ours**. This similarity may be attributed to the fact that Eq. (6) in our paper can be interpreted as a subset-based approximation of the Shapley value. Nevertheless, the Shapley value is significantly more **computationally expensive**.
>
> ||Attention|Shapley value|Ours|
> |-|-|-|-|
> |NL|91.3|100.0|100.0|
>
> > Reference
>
> Thank you for pointing them out. We will include citations for SCM and the BERT paper in the updated version.
>
> > Notations, typos and suggestions
>
> We appreciate the detailed feedback. We will improve the clear definitions of all notations in (A.6) and (A.11), fix the typos in (A.7), and update Eq.(6) to use a single vertical line for denoting absolute value, as suggested.
>
> Thank you again for your valuable and encouraging feedback. We will incorporate all these clarifications and improvements into our revision.

---

> > ### Comment · Reviewer_igVU · 2025-04-04
> >
> > Thank you for your response. I have updated my recommendation accordingly.

---

> > > ### Author Response · Authors · 2025-04-04
> > >
> > > We appreciate your comments and positive response. Thank you for all the time and efforts in reviewing this paper.

---

### Official Review · Reviewer_rJvN · 2025-03-16

**Overall Recommendation:** 4

**Summary:**

This paper proposes CoSo, a reinforcement learning approach for finetuning the VLM agent. The theoretical analysis shows that CoSo can guarantee the property of convergence and performance. Experimental results demonstrate that CoSo achieves superior performance on a range of control tasks.

**Claims And Evidence:**

The authors define the metric (Eq. (4) and (6)) to measure the effect of the token $y^i$ for a given action $a$ which is the likelihood difference between $y$ and $y^{-i} \cup y^{i}_{null}$. Later on, the casual-weighted entropy is defined in Eq. (7), the weighted term aims to measure one token $y^i$ and all tokens $y$ instead of the corresponding previous tokens $y^{1:i-1}$. To the reviewer, the weighted term $\mathcal{B}^i\_{y\rightarrow a}$ does not align with the entropy term $\mathcal{H}(y^i|y^{1:i-1})$.

**Essential References Not Discussed:**

Some related papers can be considered for discussion, but it is not compulsory.

[1] Bai, Hao, et al. "Digi-Q: Learning Q-Value Functions for Training Device-Control Agents." arXiv preprint arXiv:2502.15760 (2025).

[2] Wang, Taiyi, et al. "Distrl: An asynchronous distributed reinforcement learning framework for on-device control agents." arXiv preprint arXiv:2410.14803 (2024).

[3] Wu, Qingyuan, et al. "VSC-RL: Advancing Autonomous Vision-Language Agents with Variational Subgoal-Conditioned Reinforcement Learning." arXiv preprint arXiv:2502.07949 (2025).

**Experimental Designs Or Analyses:**

I have checked the soundness and validity of the experimental designs and analyses.

**Methods And Evaluation Criteria:**

To evaluate the efficiency of the proposed approaches, this paper adopts AitW, Gym Cards and ALFWorld which are commonly-used and challenging benchmarks.

**Other Comments Or Suggestions:**

None.

**Other Strengths And Weaknesses:**

None.

**Questions For Authors:**

To the reviewer, the weighted term $\mathcal{B}^i\_{y\rightarrow a}$ does not align with the entropy term $\mathcal{H}(y^i|y^{1:i-1})$.

**Relation To Broader Scientific Literature:**

This paper provides a novel RL method to enhance the learning efficiency of VLM agents in addressing challenging control tasks (e.g., AitW [1]).
CoSo is an interesting RL method based on counterfactual reasoning, and it has the potential to be applied to address other challenging control tasks.


[1] Rawles, Christopher, et al. "Androidinthewild: A large-scale dataset for android device control." Advances in Neural Information Processing Systems 36 (2023): 59708-59728.

**Theoretical Claims:**

I have checked the correctness of Eq. (2), Lemma 4.2, Lemma 4.3 and Proposition 4.4.

---

> ### Author Rebuttal · Authors · 2025-03-31
>
> We sincerely appreciate your valuable comments and the time you took to review our work. We also appreciate your positive remarks about the potential applicability of our method to other challenging control tasks. Regarding your questions, we provide our responses below:
>
> > Q1.  The weighted term and the entropy term
>
> - Thanks for the question. $\mathcal{B}^{i}_{y \rightarrow a}$ denotes the causal weight of token $y^i$, while $\mathcal{H}(y^i | y^{1:i-1})$ denotes the conditional entropy of the same token. They are therefore **aligned** in the objective.
>
> - Moreover, the causal weight $\mathcal{B}_{y\rightarrow a}^i$ is **used only as a scalar factor** (without gradient) for the entropy term $\mathcal{H}(y^i | y^{1:i-1})$. As such, the exact computation of the causal weight is flexible, and in practice, it can be implemented in different ways—globally, locally, or even based on random subsets of $y^{1:n}$ like Shapley Value (as mentioned by Reviewer igVU).
>
> > Reference
>
> Thanks for the helpful suggestions. We will add the recommended references and include a discussion of them in the updated Related Work section.

---

> > ### Comment · Reviewer_rJvN · 2025-04-01
> >
> > Thanks for the authors' response, and my concerns have been addressed.

---

> > > ### Author Response · Authors · 2025-04-02
> > >
> > > We appreciate the reviewer's comments and positive response. We thank the reviewer for all the time and efforts in reviewing this paper.

---

### Decision · Program_Chairs · 2025-05-01

**Decision:**

Accept (poster)

**Comment:**

This work proposes an RL framework for finetuning the VLM agent.   All reviewers consistently recommended accepting this work. AC agrees that this work is interesting and deserves to be published on ICML 2O25. The reviewers did raise some valuable concerns that should be addressed in the final camera-ready version of the paper.